# ALIGNING LLMs WITH GRAPH NEURAL SOLVERS FOR COMBINATORIAL OPTIMIZATION

## ABSTRACT

Recent research has demonstrated the effectiveness of large language models (LLMs) in solving combinatorial optimization problems (COPs) by representing tasks and instances in natural language. However, purely language-based approaches struggle to accurately capture complex relational structures inherent in many COPs, rendering them less effective at addressing medium-sized or larger instances. To address these limitations, we propose AlignOPT, a novel approach that aligns LLMs with graph neural solvers to learn a more generalizable neural COP heuristic. Specifically, AlignOPT leverages the semantic understanding capabilities of LLMs to encode textual descriptions of COPs and their instances, while concurrently exploiting graph neural solvers to explicitly model the underlying graph structures of COP instances. Our approach facilitates a robust integration and alignment between linguistic semantics and structural representations, enabling more accurate and scalable COP solutions. Experimental results demonstrate that AlignOPT achieves state-of-the-art results across diverse COPs, underscoring its effectiveness in aligning semantic and structural representations. In particular, AlignOPT demonstrates strong generalization, effectively extending to previously unseen COP instances.

## INTRODUCTION

Combinatorial optimization problems (COPs), which involve finding optimal solutions from finite sets of objects, underpin numerous real-world applications in logistics, scheduling, and network design (Bengio et al., 2021). Typical COPs, such as the Traveling Salesman Problem (TSP), Vehicle Routing Problem (VRP), and Knapsack Problem (KP), are notoriously challenging due to their NP-hard nature, requiring efficient heuristic or meta-heuristic solutions (Wang & Sheu, 2019; Konstantakopoulos et al., 2022; Lin et al., 2024). Traditionally, COPs have been approached through exact optimization methods and domain-specific heuristics. However, these methods often require extensive domain knowledge and manual tuning, making them less adaptable to new problem variants or different application contexts.

Recent studies indicate that large language models (LLMs) have emerged as powerful and versatile tools for tackling a diverse range of COPs. By framing COPs within natural language descriptions, LLM-based methods have demonstrated initial success in automatically solving optimization problems. Nevertheless, despite these advancements, the current capability of LLMs to directly generate solutions remains primarily restricted to relatively small-scale problem instances, such as TSP with fewer than 30 nodes (Yang et al., 2023; Iklassov et al., 2024). In addition, existing LLM-based solutions still encounter inherent limitations when addressing COPs characterized by complex underlying structures, particularly graph problems (Cappart et al., 2023; Bengio et al., 2021; Drakulic et al., 2024). Pure language models inherently lack explicit structural reasoning capabilities, making it difficult for them to effectively capture and represent intricate relational information in graphs. Consequently, these limitations can significantly degrade solution optimality and overall quality, substantially limiting the applicability of LLM-driven approaches in realistic, large-scale settings, particularly in fields such as logistics, transportation, and supply chain management, where typical problem instances involve hundreds to thousands of nodes (Bengio et al., 2021).

To address these challenges, we propose AlignOPT, a novel framework designed to integrate the complementary capabilities of LLMs and graph-based neural solvers for COPs. Specifically, LLMs

provide robust semantic understanding and flexible representation of natural language instructions, while graph-based neural solvers explicitly capture relational structures and topological dependencies inherent in COP instances. To effectively align these two modalities, AlignOPT introduces a multi-task pre-training strategy comprising two novel objectives: (1) a Text-Graph Contrastive (TGC) loss, designed to align semantic node embeddings from LLMs with structural embeddings from graph-based neural solvers, and (2) a Text-Graph Matching (TGM) loss, facilitating fine-grained multimodal node representation. By jointly optimizing these objectives, AlignOPT produces unified representations that enhance the accuracy and richness of COP embeddings. In this way, AlignOPT leverages guidance from LLMs exclusively during the pre-training stage to embed optimization knowledge into the graph neural solver (encoder). In the fine-tuning stage, AlignOPT fine-tunes the graph encoder along with a decoder trained via reinforcement learning to learn effective optimization policy. Consequently, AlignOPT utilizes only the graph encoder and decoder for inference, processing inputs directly as graphs without relying on textual input or an LLM. This approach significantly reduces inference overhead and enhances computational efficiency, enabling AlignOPT to achieve superior generalization and solution quality across diverse COPs.

Overall, the main contributions of this work to the COPs research can be summarized as follows.

- We introduce a novel framework AlignOPT, that explicitly **aligns LLMs with graph-based neural solvers**, bridging the gap between semantic and structural representations in COPs and moving beyond the single-modality reliance of current LLM-based models.
- AlignOPT performs **multi-task pre-training across diverse text-attributed COPs**, facilitating a more informative encoding process and subsequent fine-tuning. This enables the generation of effective and unified solutions for various COPs and adapts efficiently to unseen COPs without further reliance on LLMs during inference.
- Extensive experiments on synthetic COP instances and real-world benchmarks demonstrate the effectiveness of our proposed AlignOPT, achieving high performance gains over state-of-the-art solvers.

## RELATED WORK

**Neural Combinatorial Optimization**   Constructive neural combinatorial optimization (NCO) methods aim to learn policies that iteratively construct solutions in an autoregressive manner. Early approaches primarily employed pointer networks (Vinyals et al., 2015; Bello et al., 2016), a class of recurrent neural networks (RNNs) that encode inputs and generate outputs through a sequence-to-sequence framework. Building on the Transformer architecture (Vaswani et al., 2017), the Attention Model (AM) (Kool et al., 2018) was subsequently developed to address vehicle routing problems (VRPs), demonstrating superior performance compared to traditional heuristic methods. Following this, various strategies have been proposed to further improve Transformer-based NCO models by exploiting the inherent symmetries in combinatorial optimization problems (COPs) (Kwon et al., 2020; Kim et al., 2022; Fang et al., 2024) and incorporating efficient active search techniques (Hottung et al., 2021; Choo et al., 2022; Qiu et al., 2022). More recently, some work extends constructive NCO to be one-for-all solvers aiming at multiple COPs by a single model (Zhou et al., 2024; Zheng et al., 2024; Berto et al.; Drakulic et al., 2024; Li et al.). However, they are constrained by specific problem structures, such as vehicle routing, which limits their representational scope and undermines the model's learning capacity. In contrast, our AlignOPT delves into general text-attributed COPs described in natural language. Leveraging the unified semantic representations inherent in LLMs, AlignOPT enables a general model to accommodate a wide range of COPs. Compared with GOAL (Drakulic et al., 2024) which proposes a unified encoder that is trained with supervised fine-tuning. AlignOPT goes further by 1). Explicitly aligning this encoder with structured optimization insights derived from LLMs during pre-training. 2) Perform multi-task fine-tuning with reinforcement learning, ensuring superior generalization across diverse routing tasks during the fine-tuning stage. These enhancements explicitly encode generalized optimization reasoning from LLMs, enabling the model to robustly generalize to diverse routing problems encountered in practice.

**LLM for Combinatorial Optimization**   Recent research on the application of LLMs to COPs has demonstrated promising and impactful results. As early attempts, LLMs operate as black-box solvers that either directly generate feasible solutions with natural language problem descriptions

(Abgaryan et al., 2024) or iteratively refine initial solutions through guided search mechanisms (Yang et al., 2023; Liu et al., 2024b). Notably, recent findings indicate that LLMs often exhibit limited generalization capabilities, tending instead to replicate memorized patterns from training data rather than performing robust, adaptable reasoning (Zhang et al., 2024; Iklassov et al., 2024). On the other hand, LLMs can be tasked with generating executable code that implements heuristic algorithms for solving COPs (Romera-Paredes et al., 2024; Liu et al., 2024a; Ye et al., 2024). By initializing a code template, LLMs iteratively refine algorithmic heuristics through an evolutionary process. While this approach demonstrates promising flexibility, it often requires substantial domain expertise and incurs high token usage for each specific problem instance. The most relevant work to us is LNCS (Jiang et al., 2024), which integrates LLMs with NCO model to unify the solution process across multiple COPs. However, LNCS sequentially utilizes LLMs and Transformer architectures, resulting in a notable modality gap when compared to specialized neural solvers designed explicitly for COPs. Moreover, LNCS heavily depends on the inference efficiency of LLMs, which is frequently constrained by significant computational requirements and limited context lengths, thus restricting their scalability when inference on large-scale COPS. Instead, we propose AlignOPT to align LLMs, adept at semantic understanding, with graph-based neural solvers, proficient in capturing structural information, aiming to enhance solution quality and generalization capabilities. Note that after pre-training of AlignOPT, LLMs are no longer required during the fine-tuning and inference stages. This allows inference to be performed rapidly without the latency or cost associated with real-time LLM queries, significantly enhancing practical usability, scalability, and deployment feasibility.

## PRELIMINARIES

**Combinatorial Optimization Problems** Solving COPs involves identifying the optimal solution from a finite set of feasible candidates. Such problems are defined by their discrete nature, with solutions commonly represented as integers, sets, graphs, or sequences (Blum & Roli, 2003). Most COPs can be defined over a graph $\mathcal{G}$ with nodes and edges. Specifically, a COP $P = (S, f)$ can be formulated as follows:

$$\min_X f(X, P) \quad \text{s.t.} \quad c_j(X, P) \leq 0, \ j = 0, 1, \dots, J. \tag{1}$$

where $X = \{x_i \in D_i \mid i = 1, \dots, n\}$ is a set of discrete variables; $f(X, P)$ indicates the objective function of COP and $c(X, P)$ denotes the problem-specific constraints for the variable $X$. Note that typical COPs (e.g., TSP, CVRP, KP) are NP-hard problems. As a result, identifying the optimal solution $s^*$ is computationally intractable in many practical scenarios. Therefore, a more tractable approach involves searching for a set of feasible solutions $S$ rather than striving for exact optimality. The set $S$ is formally defined as:

$$S = \{s = \{(x_1, v_1), \dots, (x_n, v_n)\} \mid v_i \in D_i, \ c(X, P) \leq 0\}. \tag{2}$$

where a solution $s$ satisfies $f(s, P) \geq f(s^*, P), \forall s \in S$.

**Neural Construction Heuristics for COPs** Learning construction heuristics has become a widely adopted paradigm for addressing Vehicle Routing Problems (VRPs) (Bello et al., 2016; Kool et al., 2018; Kwon et al., 2020). In this framework, solutions are constructed incrementally by sequentially selecting valid nodes, a process effectively modeled as a Markov Decision Process (MDP). At each step, the agent observes a state composed of the problem instance and the current partial solution, and selects a valid node from the remaining candidates. This process continues iteratively until a complete and feasible solution is obtained.

The solution construction policy is typically parameterized by a neural network, such as a Long Short-Term Memory (LSTM) or Transformer, denoted by $\theta$. At each decision step, the policy infers a probability distribution over the valid nodes, from which one is sampled and appended to the partial solution. The overall probability of generating a tour $\pi$ is then factorized as $p_\theta(\pi|\mathcal{G}) = \prod_{t=1}^{T} p_\theta(\pi_t|\mathcal{G}, \pi_{<t})$, where $\pi_t$ denotes the node selected at time step $t$, and $\pi_{<t}$ represents the sequence of previously selected nodes (i.e., the current partial solution). To optimize the policy parameters $\theta$, the REINFORCE algorithm (Williams, 1992), a foundational policy gradient method in deep reinforcement learning, is commonly utilized.

$$\nabla_\theta \mathcal{L}(\theta|\mathcal{G}) = \mathbb{E}_{p_\theta(\pi|\mathcal{G})}[(c(\pi) - b(\mathcal{G}))\nabla \log p_\theta(\pi|\mathcal{G})]. \tag{3}$$

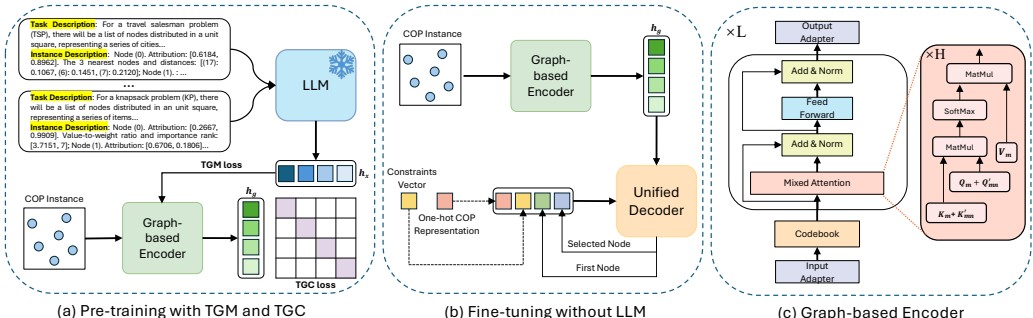

(a) Pre-training with TGM and TGC    (b) Fine-tuning without LLM    (c) Graph-based Encoder

Figure 1: Overall workflow of AlignOPT. (a) AlignOPT first performs multi-task pretraining on diverse COPs to align semantic and structural node representations with TGC and TGM losses. The LLM remains frozen and processes the TAIs to generate semantic node representations. (b) The encoder and decoder are then fine-tuned through reinforcement learning to solve COPs. Notably, LLMs are excluded during this phase to ensure computational efficiency, as the encoder has already been aligned with LLM-derived representations during pre-training. (c) The model architecture of the graph-based encoder, which applies a mixed attention mechanism that enables handling COPs represented by graphs.

where $c(\pi)$ is the cost of the constructed tour $\pi$ (e.g., total length), and $b(\cdot)$ is an action-independent baseline function employed to reduce the variance of the gradient estimates.

## THE PROPOSED FRAMEWORK

We propose AlignOPT, a unified framework to align LLMs with graph-based neural solvers for solving COPs. The overall framework of AlignOPT is illustrated in Fig. 1. This section first describes how AlignOPT derives node representations from LLMs and graph-based encoders, followed by detailing its pre-training objectives.

### COP-SPECIFIC TEXT-ATTRIBUTED REPRESENTATIONS

We start from a recent work LNCS, which represents each COP instance as a text-attributed instance (TAI) (Jiang et al., 2024). Specifically, the COPs are denoted by $\mathcal{T}(\mathcal{G}^P) = \{\kappa^P, v^P\}$, where $\kappa^P$ is the task description specifying the general structure of the problem, such as decision variables, constraints, and objective function, while $v^P$ is the instance description detailing node- or edge-specific features. Specifically, both the instance and the task description are encoded by the LLM, denoted by $x_i^P = \text{LLM}(v_i^P)$ and $k^P = \text{LLM}(\kappa^P)$, respectively. The resulting node embeddings $\{x_i^P\}_{i=1}^n$ encapsulate information specific to each instance, whereas the task embedding $k^P$ captures domain-specific semantic attributes pertinent to the COP $P$. In this work, AlignOPT incorporates task representation $k^P$ into the LLM pathway to obtain COP-specific text-attributed representations. Each node's LLM representation is enhanced with its task representations (i.e., $x'^P_i = \text{Concat}\left(x_i^P, k^P\right)$).

While this design verifies that neural solvers can be enhanced by the semantics representation of COPs with LLMs, the semantic and structural modalities of COPs remain loosely coupled. In the following subsection, we present how AlignOPT addresses this limitation by: (1) modeling COPs with a graph-based neural encoder that captures the structural dependencies among nodes; and (2) pretraining the solver on a diverse set of COP instances while aligning its representations with those of an LLM through a contrastive loss objective.

### GRAPH-BASED NEURAL ENCODER

We apply a graph-based neural encoder in AlignOPT, capturing the structural node representations that inherently exist in COPs. Specifically, the encoder stems from the architecture of GOAL (Drakulic et al., 2024), which employs a backbone comprising shared self-attention transformer layers alongside task-specific adapter modules for learning a generalist solver. Specifically, the back-

bone architecture includes (1) *task-specific low-rank adapter* modules for input and output processing, (2) a *shared codebook* that projects low-dimensional node/edge features into the full hidden space, (3) a stack of *shared mixed attention blocks*. Keeping the same use of the first two parts, we detail how we structure the mixed attention to extend standard self-attention for integrating node and edge components in attention scores.

Instead of attention scores solely computed with node representations, for each mixed-attention head $h$, node representations are linearly projected into query ($Q_n^{(h)}$), key ($K_m^{(h)}$), and value ($V_m^{(h)}$) vectors, while edge representations $E_{mn}$ are projected separately into corresponding query-like ($Q_{mn}'^{(h)}$) and key-like ($K_{mn}'^{(h)}$) vectors as follows:

$$K'^{(h)}_{mn} = E_{mn}W_K'^{(h)} \quad Q'^{(h)}_{mn} = E_{mn}W_Q'^{(h)}. \tag{4}$$

Consequently, the attention score is computed as:

$$S_{mn}^{(h)} = \langle K_m^{(h)} + K_{mn}'^{(h)} | Q_n^{(h)} + Q_{mn}'^{(h)} \rangle. \tag{5}$$

where the inner product $\langle | \rangle$ adds node and edge representations and calculates the attention scores by standard self-attention (Vaswani et al., 2017). The resulting attention scores computed across all attention heads are subsequently processed by applying an optional log-binary mask $\mathcal{M}$, This ensures that attention is only computed between node-edge pairs that satisfy both the task-specific feasibility criteria (i.e., valid interactions required by the combinatorial optimization task) and graph structural constraints (i.e., connections reflecting the underlying graph topology). Following this masking step, the scores undergo column-wise softmax normalization, yielding the final normalized attention distributions. Consequently, the final output representation of mixed attention $\{g_i^P\}_{i=1}^N$ of the $N$ input query nodes is an $g \in \mathbb{R}^{N \times d_g}$ matrix:

$$\mathbf{g^P} = \sum_h \text{softmax}_{\text{col}}(S_{mn}^{(h)} + \mathcal{M})^\top V_m^{(h)} W_O^{(h)\top}. \tag{6}$$

To ensure dimensional compatibility with LLM-generated semantic representations, both textual representations and graph-based representations are collected through a comprehensive encoding pipeline. Specifically, the textual representations $\mathbf{x}^P \in \mathbb{R}^{N \times d_l}$ are obtained by processing node-level natural language descriptions using frozen LLMs (e.g., Llama3.1 8B). Specifically, tokenized descriptions are encoded into embeddings $\mathbf{E}_{\text{node}} \in \mathbb{R}^{N \times S \times D}$ and mean-pooled over tokens to form compact node embeddings $\mathbf{h}_i \in \mathbb{R}^D$, capturing semantic information from problem formulations. The graph-based representations $\mathbf{g}^P \in \mathbb{R}^{N \times d_g}$ are derived via graph encoders, employing message-passing operations on problem-specific graphs to encode structural dependencies and topological constraints consistent with downstream COPs. Both representations (i.e., $\mathbf{x^P}$ and $\mathbf{g^P}$) are then linearly projected into a unified latent space, resulting in LLM representations $\mathbf{h_x} \in \mathbb{R}^{N \times d_h}$ and graph representations $\mathbf{h_g} \in \mathbb{R}^{N \times d_h}$ for each COP instance.

### ALIGNING LLM WITH GRAPH-BASED NEURAL SOLVERS

While the graph-based encoder captures structural patterns of COPs, LLMs encode semantic aspects, such as textual objectives, constraints, and heuristic rules. Aligning these representations enables integrated structural-semantic reasoning, enhancing solution quality and generalization. To this end, we introduce two pre-training objectives: a text-graph contrastive (TGC) loss that aligns semantic and structural node representations, and a text-graph matching (TGM) loss that facilitates fine-grained multimodal node embeddings.

**Text-Graph Contrastive (TGC) Loss** Inspired by recent advances in vision-language contrastive paradigms (Chen et al., 2020; Li et al., 2022), AlignOPT extends the InfoNCE loss to bridge the modality gap between textual and graph-based representations for solving COPs. Positive pairs comprise LLM and graph representations of identical nodes, whereas negative pairs include embeddings from distinct nodes within the same batch. The proposed text-graph contrastive (TGC) loss maximizes positive pair similarity and minimizes negative pair similarity:

$$\mathcal{L}_{\text{TGC}} = -\log \frac{\exp\left(sim(\mathbf{h}_x^i, \mathbf{h}_g^i)/\tau\right)}{\sum_{j=1}^B \mathbb{1}_{[j \neq i]} \exp\left(sim(\mathbf{h}_x^i, \mathbf{h}_g^j)/\tau\right)}. \tag{7}$$

where $\mathbf{h}_x^i$ and $\mathbf{h}_g^i$ are LLM and graph representations of node $i$ retrieved from $\mathbf{h_x} \in \mathbb{R}^{N \times d_h}$ and $\mathbf{h_g} \in \mathbb{R}^{N \times d_h}$, $sim(\cdot, \cdot)$ denotes the cosine similarity function, $\tau$ is a temperature hyperparameter scaling similarity scores, and $B$ represents the batch size.

**Text-Graph Matching (TGM) Loss**  In addition to the TGC loss, which aligns the textual node representations and graph-based node representations in a shared latent space, we further introduce a Text-Graph Matching (TGM) objective, which is formulated as a binary classification task that encourages the model to explicitly distinguish between positive (matched) or negative (unmatched) text-graph pairs. Specifically, each graph-based representation $\bar{\mathbf{h}}_{\mathbf{g}} = \frac{1}{N} \sum_i \mathbf{h}_g^i$ is paired with two types of textual features: positive textual features $\bar{\mathbf{h}}_{\mathbf{x}} = \frac{1}{N} \sum_i \mathbf{h}_x^i$ from the identical problem instance, and negative textual features randomly sampled from other instances within the same batch. The ground truth labels are constructed automatically based on instance correspondence: a pair $(\bar{\mathbf{h}}_{\mathbf{x_i}}, \bar{\mathbf{h}}_{\mathbf{g_j}})$ is labeled as positive $(y = 1)$ if $j = i$, and negative $(y = 0)$ otherwise. The concatenated vector $[\bar{\mathbf{h}}_{\mathbf{x_i}}, \bar{\mathbf{h}}_{\mathbf{g_j}}]$ is fed into a binary classification head to predict the matching probability:

$$p_{ij} = \sigma \left( \text{MLP}([\bar{\mathbf{h}}_{\mathbf{x_i}}, \bar{\mathbf{h}}_{\mathbf{g_j}}]) \right), \tag{8}$$

where $\sigma$ is the sigmoid function. The TGM loss is then defined as the binary cross-entropy:

$$\mathcal{L}_{\text{TGM}} = -\frac{1}{M} \sum_{i=1}^{M} \sum_{j=1}^{M} \left[ y_{ij} \log p_{ij} + (1 - y_{ij}) \log(1 - p_{ij}) \right], \tag{9}$$

where $M$ is the batch size, and $y_{ij} = \mathbb{1}_{[j=i]}$ is the ground truth label indicating whether the text and graph representations originate from the same instance. A textual representation is considered to be noisy if the TGM head predicts it as unmatched to the graph-based representation. The overall training objective is:

$$\mathcal{L} = \mathcal{L}_{\text{TGC}} + \lambda \cdot \mathcal{L}_{\text{TGM}}, \tag{10}$$

where $\lambda$ is a task-balancing coefficient. This dual-loss framework explicitly encourages fine-grained alignment between textual semantics and structural graph embeddings, enhancing robustness against modality misalignment and improving generalization to diverse combinatorial optimization instances. We provide an ablation study to investigate the effectiveness of the joint loss functions in Table 3.

### FINE-TUNING SCHEMES

After pretraining the model to align textual (LLM-derived) and structural (graph-derived) representations, AlignOPT employs two distinct fine-tuning paradigms, both leveraging a unified decoder trained via reinforcement learning. *Single-Task Fine-Tuning (STFT)* optimizes model parameters using data exclusively from every single COP. *Multi-Task Fine-Tuning (MTFT)* simultaneously trains on diverse COPs, using a stochastic sampler that constructs batches by selecting $p\%$ ($p \sim \mathcal{U}(30, 50)$) samples from a single randomly chosen task and the remaining $(100 - p)\%$ uniformly from other tasks. AlignOPT follows existing works to utilize a multi-head self-attention based decoder to generate COP solutions (Kool et al., 2018). The model is then trained with a conflict-free reinforcement learning for multi-task training for COPs (Jiang et al., 2024).

### EXPERIMENTS

**Experimental Settings**  The proposed AlignOPT is evaluated across five representative COPs: the Traveling Salesman Problem (TSP), Capacitated Vehicle Routing Problem (CVRP), Knapsack Problem (KP), Minimum Vertex Cover Problem (MVCP), and Single-Machine Total Weighted Tardiness Problem (SMTWTP). Additionally, the pre-trained AlignOPT is fine-tuned on two unseen tasks, including the Vehicle Routing Problem with Backhauls (VRPB) and the Maximum Independent Set Problem (MISP). The evaluation leverages synthetic COP instances, with detailed procedures for data generation and their corresponding TAI examples provided in the supplementary materials.

**Baselines**  We compare our AlignOPT with LLM-based solvers, traditional solvers, and NCO solvers. **(1) LLM-based Solvers:** We begin by comparing our approach with existing LLM-based

| | Method | $n = 20$ | $n = 50$ | $n = 100$ |
|---|---|---|---|---|
| | AEL | 7.78% | 10.50% | 12.35% |
| | ReEvo | 7.77% | 10.23% | 11.87% |
| | SGE | 11.32% | 45.28% | - |
| _TSP_ | LMEA* | 3.94% | - | - |
| | ORPO* | 4.40% | 133.0% | - |
| | LNCS | 0.39% | 1.62% | 4.38% |
| | AlignOPT(MTFT) | 0.00% | 0.53% | 1.03% |
| | **AlignOPT(STFT)** | **0.00%** | **0.35%** | **0.38%** |
| | ReEvo | 5.19% | 14.27% | 19.59% |
| | SGE | 76.46% | 144.21% | - |
| _CVRP_ | LNCS | 2.54% | 3.63% | 5.58% |
| | AlignOPT(MTFT) | 1.31% | 3.47% | 5.05% |
| | **AlignOPT(STFT)** | **0.49%** | **3.09%** | **4.39%** |
| | ReEvo | 0.14% | 4.31% | 9.40% |
| | SGE | 42.62% | 39.08% | - |
| _KP_ | LNCS | 0.10% | 0.07% | 0.04% |
| | AlignOPT(MTFT) | 0.08% | 0.03% | 0.12% |
| | **AlignOPT(STFT)** | **0.00%** | **0.00%** | **0.00%** |

Table 1: The optimality gaps of LLM-based approaches on different tasks. *: Results are drawn from the original literature. -: Excessively long time leads to unavailability. Bold indicates the best results among comparable methods.

methods, including **OPRO** (Yang et al., 2023) and **LMEA** that aim to directly generate solutions from textual descriptions of the optimization problems. We further consider (Liu et al., 2024b), **AEL** (Liu et al., 2023), **ReEvo** (Ye et al., 2024), and **SGE** (Iklassov et al., 2024), which leverage LLMs to autonomously generate heuristic strategies for solving COPs. Specifically, AEL and ReEvo are applied to evolve constructive heuristics for the TSP, while ReEvo is also employed to enhance the ant colony optimization (ACO) method for solving the CVRP and the KP. **(2) Traditional Solvers:** We utilize **OR-Tools**, a heuristic optimization framework, to address the TSP, CVRP, and KP. In addition, we benchmark against established heuristic methods, including the **nearest neighbor** and **farthest insertion** heuristics for TSP; the **sweep** algorithm and the **parallel savings** algorithm for CVRP (Rasku et al., 2019); a **greedy policy** for KP; the **MVCApprox** method (Bar-Yehuda & Even, 1985) and the **REH** (Pitt, 1985) for MVCP; and **EDD** dispatching rule (Jackson, 1955) for SMTWTP. We also include Ant Colony Optimization (ACO) as a metaheuristic baseline, configured with 20 ants and 50 iterations (Ye et al., 2023). **(3) NCO Solvers:** Since AlighOPT aims at a wide spectrum of COPs, we compare it with **GOAL** (Drakulic et al., 2024), the state-of-the-art one-for-all solver trained with supervised learning for assorted COPs. Likewise, we compare with **LNCS** (Jiang et al., 2024), a LLM-based NCO solver that addressed disparate COPs.

**Comparison with LLM-based Solutions**  The experimental comparison presented in Table 1 evaluates the performance of our proposed AlignOPT method against recent LLM-based methods across 3 representative COPs. To be specific, AlignOPT(STFT) consistently achieves the lowest optimality gaps across TSP, CVRP, and KP, significantly outperforming other recent LLM-based methods such as AEL, ReEvo, SGE, LMEA, and ORPO. For instance, in TSP, AlignOPT(STFT) attains gaps of only 0.00%, 0.35%, and 0.38% at problem sizes 20, 50, and 100, respectively, markedly better than LNCS (0.39%, 1.62%, 4.38%) and competitors like ReEvo and SGE, which exhibit gaps exceeding 10% at larger sizes. In CVRP, AlignOPT(STFT) demonstrates significantly smaller gaps (0.49%, 3.09%, and 4.39% respectively), substantially outperforming methods like ReEvo and SGE, which present notably higher gaps, especially at larger instances. For KP, AlignOPT(STFT) achieves perfect optimality (0.00% gap) across all evaluated sizes, clearly surpassing the performance of LNCS (0.10%, 0.07%, 0.04%), ReEvo (up to 9.40% on $n = 100$), and SGE (up to 42.62% on $n = 20$). These results validate the effectiveness of AlignOPT in solving relatively large COPs (i.e., $n > 30$) by leveraging the structural information inherently embedded in their formulations.

**Comparison with Traditional and NCO solvers**  We present the experimental comparison between AlignOPT and baselines in Table 2. Overall, AlignOPT consistently achieves competitive performance across various problem sizes ($n = 20, 50, 100$). Specifically, AlignOPT(STFT), which fine-tunes on task-specific instances, demonstrates superior or comparable results to all baseline methods. For instance, in TSP, AlignOPT(STFT) achieves the lowest objective values at all sizes, closely matching the state-of-the-art solver LKH3 and significantly outperforming classical heuris-

| | Method | $n = 20$ | | | $n = 50$ | | | $n = 100$ | | |
| --- | --- | --- | --- | --- | --- | --- | --- | --- | --- | --- |
| | | Obj. | Gap | Time | Obj. | Gap | Time | Obj. | Gap | Time |
| TSP | LKH3 | 3.85 | 0.00% | 0.05s | 5.69 | 2.80% | 0.26s | 7.76 | 0.00% | 2.05s |
| | OR tools | 3.85 | 0.00% | 0.36s | 5.87 | 3.07% | 0.60s | 8.13 | 4.77% | 1.32s |
| | Nearest neighbor | 3.91 | 1.45% | 0.06s | 5.89 | 3.51% | 0.03s | 9.69 | 24.87% | 0.10s |
| | Farthest insertion | 3.96 | 2.89% | 0.21s | 5.98 | 4.97% | 4.73s | 8.21 | 5.80% | 126s |
| | ACO | 3.94 | 2.23% | 0.74s | 6.54 | 14.54% | 1.53s | 9.99 | 28.74% | 2.01s |
| | LNCS | 3.87 | 0.55% | 0.31s | 5.79 | 1.64% | 0.49s | 8.10 | 4.38% | 0.81s |
| | GOAL | 3.86 | 0.26% | 0.012s | 5.76 | 1.23% | 0.018s | 7.98 | 2.84% | 0.028s |
| | AlignOPT(MTFT) | 3.85 | 0.00% | 0.048s | 5.74 | 0.53% | 0.082s | 7.84 | 1.03% | 0.165s |
| | AlignOPT(STFT) | **3.85** | 0.00% | 0.048s | **5.71** | 0.35% | 0.082s | **7.79** | 0.38% | 0.165s |
| CVRP | HGS | 6.10 | 0.00% | 0.2s | 10.36 | 0.00% | 0.6s | 15.49 | 0.00% | 2.22s |
| | OR tools | 6.18 | 1.30% | 0.27s | 11.05 | 6.63% | 0.48s | 17.36 | 12.07% | 1.40s |
| | Sweep heuristic | 7.51 | 23.17% | 0.01s | 15.65 | 50.95% | 0.05s | 28.40 | 83.39% | 0.25s |
| | Parallel saving | 6.33 | 3.85% | <0.01s | 10.90 | 5.18% | <0.01s | 16.42 | 6.03% | 0.03s |
| | ACO | 7.72 | 26.56% | 0.80s | 15.76 | 52.12% | 1.97s | 26.66 | 72.11% | 4.90s |
| | LNCS | 6.25 | 2.51% | 0.315s | 10.74 | 3.62% | 0.495s | 16.35 | 5.59% | 0.820s |
| | GOAL | 6.20 | 1.50% | 0.013s | 10.73 | 3.55% | 0.019s | 16.30 | 5.30% | 0.029s |
| | AlignOPT(MTFT) | 6.18 | 1.31% | 0.051s | 10.72 | 3.47% | 0.087s | 16.27 | 5.048% | 0.172s |
| | AlignOPT(STFT) | **6.13** | 0.49% | 0.051s | **10.68** | 3.09% | 0.087s | **16.17** | 4.39% | 0.172s |
| KP | OR tools | 7.948 | 0.00% | <0.01s | 20.086 | 0.00% | <0.01s | 40.377 | 0.00% | <0.01s |
| | Greedy policy | 7.894 | 0.67% | <0.01s | 20.033 | 0.26% | <0.01s | 40.328 | 0.12% | <0.01s |
| | ACO | 7.947 | 0.00% | 0.72s | 20.053 | 0.15% | 2.19s | 40.124 | 0.62% | 3.41s |
| | LNCS | 7.939 | 0.10% | 0.308s | 20.071 | 0.06% | 0.485s | 40.361 | 0.03% | 0.800s |
| | GOAL | 7.941 | 0.09% | 0.012s | 20.078 | 0.04% | 0.017s | 40.370 | 0.11% | 0.027s |
| | AlignOPT(MTFT) | 7.942 | 0.08% | 0.049s | 20.081 | 0.03% | 0.084s | 40.372 | 0.12% | 0.168s |
| | AlignOPT(STFT) | **7.948** | 0.00% | 0.049s | **20.085** | 0.00% | 0.084s | **40.380** | 0.00% | 0.168s |
| MVCP | Gurobi | 11.95 | 0.00% | <0.01s | 28.812 | 0.00% | 0.01s | 56.191 | 0.00% | 0.02s |
| | MVCApprox | 14.595 | 22.13% | <0.01s | 34.856 | 20.98% | <0.01s | 68.313 | 21.57% | <0.01s |
| | REH | 16.876 | 41.22% | <0.01s | 41.426 | 43.78% | <0.01s | 81.860 | 45.68% | <0.01s |
| | LNCS | 12.900 | 7.93% | 0.310s | 32.101 | 11.42% | 0.485s | 64.893 | 15.49% | 0.800s |
| | GOAL | 12.750 | 6.50% | 0.012s | 31.800 | 10.40% | 0.017s | 64.300 | 14.50% | 0.026s |
| | AlignOPT(MTFT) | 12.703 | 6.30% | 0.048s | 31.751 | 10.20% | 0.081s | 64.257 | 14.35% | 0.163s |
| | AlignOPT(STFT) | **12.597** | 5.41% | 0.048s | **31.562** | 9.54% | 0.081s | **64.091** | 14.06% | 0.163s |
| SMTWTP | Gurobi | 0.1017 | 0.00% | 0.02s | 0.2148 | 0.00% | <0.01s | 0.2438 | 0.00% | 0.35s |
| | ACO | 0.2967 | 191.74% | 0.35s | 1.0471 | 387.48% | 1.35s | 6.77 | 2677% | 2.00s |
| | LNCS | 0.2862 | 181.41% | 0.315s | 0.3353 | 56.10% | 0.492s | 0.3316 | 36.01% | 0.815s |
| | GOAL | 0.2848 | 179.50% | 0.013s | 0.3335 | 55.20% | 0.019s | 0.3298 | 35.20% | 0.029s |
| | AlignOPT(MTFT) | 0.2835 | 64.12% | 0.052s | 0.3328 | 35.45% | 0.089s | 0.3291 | 25.919% | 0.175s |
| | AlignOPT(STFT) | **0.2829** | 64.05% | 0.052s | **0.3318** | 35.26% | 0.089s | **0.3285** | 25.78% | 0.175s |

Table 2: Performance comparison on 1K instances. AlignOPT(MTFT) denotes multi-task fine-tuning on diverse COPs, while AlignOPT(STFT) refers to fine-tuning on the target COP. Obj. indicates the average objective values. LNCS uses LLM Encoder + Transformer Decoder, GOAL uses GNN only, and AlignOPT uses GNN + Transformer Decoder.

tics such as Nearest Neighbor and Farthest Insertion, as well as the LNCS baseline. In CVRP, AlignOPT(STFT) substantially outperforms heuristics like Sweep and Parallel Saving, delivering objective values closely aligned with HGS, the leading solver. For KP, AlignOPT(STFT) achieves optimal solutions on par with OR tools and keeps outperforming heuristic methods and LNCS. Notably, classical optimization solvers such as Gurobi consistently perform best for MVCP and SMTWTP, yet AlignOPT(STFT) significantly narrows the performance gap compared to heuristic methods and the LNCS baseline. Specifically, for MVCP at $n = 100$, AlignOPT(STFT) achieves a 14.06% gap, improving over REH (45.68%) by 31.62% and slightly outperforming LNCS (15.49%). At $n = 50$, it further reduces the gap to 9.54%, compared to REH's 43.78% and LNCS's 11.42%. For SMTWTP, where ACO struggles to produce feasible solutions across all scales, AlignOPT(STFT) consistently outperforms LNCS, achieving gaps of 25.78%, 35.26%, and 64.05% at $n = 100$, 50, and 20, respectively, compared to LNCS's 36.01%, 56.10%, and 181.41%. These results underscore AlignOPT's robust performance and its capability to generalize effectively across diverse tasks. AlignOPT (particularly STFT variant) consistently outperforms GOAL across all tested combinatorial optimization problems, while maintaining comparable computational efficiency, with STFT demonstrating superior balance between solution quality and speed.

**Generalization on Unseen COPs** Although the efficacy of AlignOPT has been validated across multiple COPs, an important consideration remains its capacity to generalize effectively to previously unseen COPs. To address this, we fine-tune the pre-trained AlignOPT model (i.e., AlignOPT(STFT)) on new COPs, specifically SDVRP, PCTSP, and SPCTSP, each with a problem size of $n = 50$. Baseline comparisons are established by randomly initializing AlignOPT and

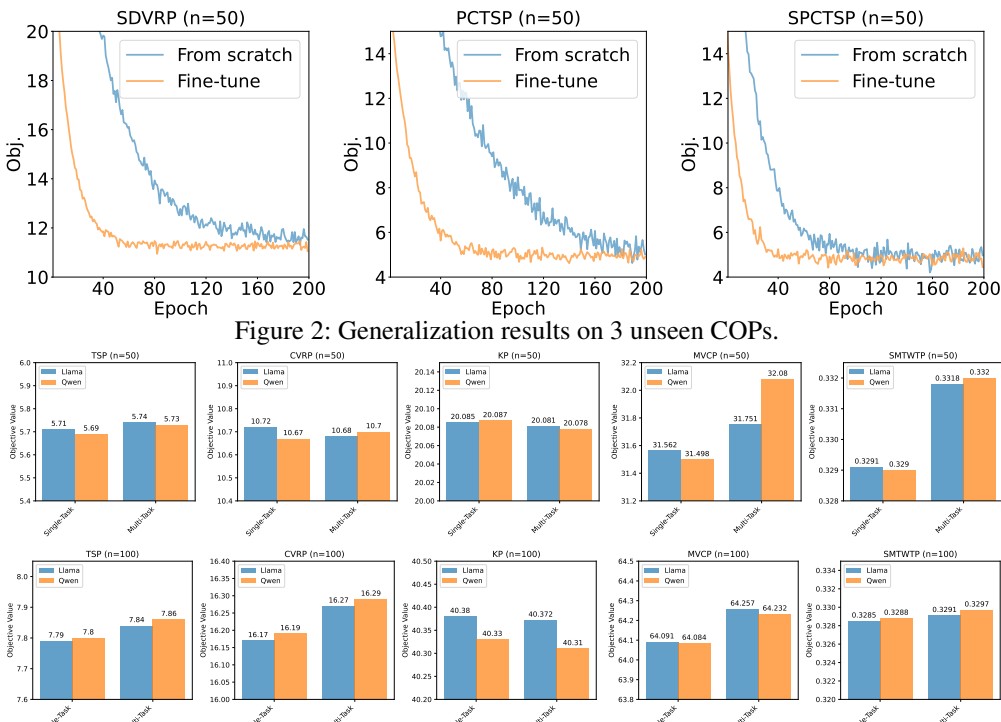

Figure 2: Generalization results on 3 unseen COPs.

Figure 3: Average Objective values of different LLMs (Llama3.1 8B and Qwen2.5 8B)

training it from scratch for 200 epochs per task. Results in Fig. 2 indicate that the pre-trained AlignOPT exhibits rapid convergence (within 40–80 epochs) and notable performance improvements, attributable to pre-learning on related routing problems (e.g., CVRP, TSP). These outcomes reinforce the generalizability of the LLM-based AlignOPT architecture and demonstrate its promise as a foundational model for diverse COPs.

## ABLATION STUDY

**Effectiveness of Key Components**   We conducted an ablation study to investigate the importance of incorporating task descriptions into node representations, and to assess the effectiveness of two proposed losses (i.e., TGC and TGM) used in the multi-task pre-training stage. To investigate the importance of LLM, we provide another variant named AlignOPT (GNS), which employs the graph encoder and decoder trained with reinforcement learning, without any LLM-derived inputs. Analysis of Table 3 yields the following insights: (1) The substantially lower performance of AlignOPT(GNS) demonstrates that structural reasoning alone (without LLM inputs) cannot account for the improvements achieved by the full model (i.e., AlignOPT(STFT)). (2) Incorporating task descriptions $k^P$ into node representations from LLMs consistently improves the model's performance. For example, on TSP with problem size 100, AlignOPT(STFT) achieved an objective value of 7.79 compared to 7.87 (w/o Task Rep.). (3) Both proposed losses, TGC and TGM, play critical roles during the pre-training stage. Specifically, removing either loss individually (w/o TGC or w/o TGM) leads to notably higher objective values and optimality gaps, such as the increase from 5.71 to 6.33 for the TGC loss ablation in TSP size 50. (4) The combined application of the above components (i.e., AlignOPT(STFT)) consistently yields the best performance across various COPs and problem sizes, underscoring the effectiveness and complementary nature of these components in AlignOPT's pre-training process. These findings collectively validate the significance of each proposed component in AlignOPT, highlighting their contributions to enhancing model performance and generalization capabilities.

**Analysis of Different LLMs**   To investigate the influence of different LLMs on AlignOPT during the pre-training stage, we conducted a comparative analysis between Llama3.1 8B and Qwen2.5 8B, focusing on problem sizes 50 and 100 under both single-task and multi-task fine-tuning scenarios. As shown in Fig. 3, Qwen slightly outperforms Llama on all five COPs at size 50 in single-task

| | Method | $n = 20$ | | | $n = 50$ | | | $n = 100$ | | |
|---|---|---|---|---|---|---|---|---|---|---|
| | | Obj. | Gap | Time | Obj. | Gap | Time | Obj. | Gap | Time |
| TSP | AlignOPT (GNS) | 4.02 | 4.41% | 0.048s | 6.33 | 11.24% | 0.082s | 8.37 | 7.86% | 0.165s |
| | AlignOPT (w/o TGC) | 3.96 | 2.86% | 0.048s | 6.18 | 8.24% | 0.082s | 8.22 | 5.52% | 0.165s |
| | AlignOPT (w/o Task Rep.) | 3.85 | 0.00% | 0.048s | 5.76 | 0.70% | 0.082s | 7.87 | 4.38% | 0.165s |
| | AlignOPT (w/o TGM) | 3.85 | 0.00% | 0.048s | 5.77 | 0.52% | 0.082s | 7.89 | 0.64% | 0.165s |
| | AlignOPT(STFT) | **3.85** | 0.00% | 0.048s | **5.71** | 0.35% | 0.082s | **7.79** | 0.38% | 0.165s |
| VRP | AlignOPT (GNS) | 6.88 | 12.79% | 0.051s | 11.21 | 8.20% | 0.087s | 17.11 | 10.46% | 0.172s |
| | AlignOPT (w/o TGC) | 6.75 | 10.12% | 0.051s | 11.05 | 6.45% | 0.087s | 16.89 | 8.24% | 0.172s |
| | AlignOPT (w/o Task Rep.) | 6.21 | 0.49% | 0.051s | 10.73 | 0.10% | 0.087s | 16.29 | 0.13% | 0.172s |
| | AlignOPT (w/o TGM) | 6.19 | 0.16% | 0.051s | 10.74 | 0.18% | 0.087s | 16.30 | 0.18% | 0.172s |
| | AlignOPT(STFT) | **6.13** | 0.49% | 0.051s | **10.68** | 3.09% | 0.087s | **16.17** | 4.39% | 0.172s |
| KP | AlignOPT (GNS) | 7.552 | 4.98% | 0.049s | 19.274 | 4.04% | 0.084s | 38.850 | 3.78% | 0.168s |
| | AlignOPT (w/o TGC) | 7.648 | 3.77% | 0.049s | 19.582 | 2.50% | 0.084s | 39.425 | 2.36% | 0.168s |
| | AlignOPT (w/o Task Rep.) | 7.941 | 0.11% | 0.049s | 20.082 | 0.01% | 0.084s | 40.375 | 0.01% | 0.168s |
| | AlignOPT (w/o TGM) | 7.942 | 0.08% | 0.049s | 20.081 | 0.02% | 0.084s | 40.372 | 0.02% | 0.168s |
| | AlignOPT(STFT) | **7.948** | 0.00% | 0.049s | **20.085** | 0.00% | 0.084s | **40.380** | 0.00% | 0.168s |
| VCP | AlignOPT (GNS) | 13.410 | 10.88% | 0.048s | 34.078 | 15.45% | 0.081s | 66.399 | 15.37% | 0.163s |
| | AlignOPT (w/o TGC) | 13.125 | 8.52% | 0.048s | 33.245 | 12.65% | 0.081s | 65.782 | 12.89% | 0.163s |
| | AlignOPT (w/o Task Rep.) | 12.741 | 0.30% | 0.048s | 31.907 | 0.49% | 0.081s | 64.438 | 0.28% | 0.163s |
| | AlignOPT (w/o TGM) | 12.731 | 0.22% | 0.048s | 31.872 | 0.38% | 0.081s | 64.398 | 0.22% | 0.163s |
| | AlignOPT(STFT) | **12.597** | 5.41% | 0.048s | **31.562** | 9.54% | 0.081s | **64.091** | 14.06% | 0.163s |
| TWTP | AlignOPT (GNS) | 0.2954 | 65.57% | 0.052s | 0.3550 | 39.49% | 0.089s | 0.3469 | 29.72% | 0.175s |
| | AlignOPT (w/o TGC) | 0.2912 | 63.25% | 0.052s | 0.3485 | 36.78% | 0.089s | 0.3412 | 27.45% | 0.175s |
| | AlignOPT (w/o Task Rep.) | 0.2843 | 0.28% | 0.052s | 0.3335 | 0.21% | 0.089s | 0.3295 | 0.12% | 0.175s |
| | AlignOPT (w/o TGM) | 0.2839 | 0.14% | 0.052s | 0.3332 | 0.12% | 0.089s | 0.3296 | 0.15% | 0.175s |
| | AlignOPT(STFT) | **0.2829** | 64.05% | 0.052s | **0.3318** | 35.26% | 0.089s | **0.3285** | 25.78% | 0.175s |

Table 3: Ablation studies of key designs across 1K instances for 5 representative COPs.

scenarios, while Llama demonstrates better performance on multi-task KP, CVRP, and SMTWTP scenarios. For the larger size 100 instances, Llama consistently achieves better results on TSP, and CVRP across both scenarios. Conversely, Qwen notably excels at MVCP and SMTWTP for both single-task and multi-task scenarios at size 100. These results suggest that model performance depends significantly on the specific COP, problem size, and fine-tuning strategy.

## CONCLUSIONS

In this work, we propose AlignOPT, a novel framework that addresses the limitations of LLM-only approaches, which struggle to accurately capture the complex relational structures of COPs. By combining the semantic understanding of LLMs with the relational modeling capabilities of graph-based neural solvers, AlignOPT effectively aligns textual descriptions with structural representations. Extensive experiments show that AlignOPT consistently achieves state-of-the-art performance. Ablation studies further validate the key design components, highlighting the effectiveness of our multi-task alignment strategy. Moreover, AlignOPT demonstrates strong generalization, successfully solving previously unseen COP instances with minimal fine-tuning and without further reliance on LLMs. Future work will focus on refining the alignment mechanisms between LLMs and graph-based solvers, particularly through dynamic integration during inference, to further enhance adaptability and performance.

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

## APPENDIX

### DATA PREPARATION

### DATA GENERATION PROCESS

To construct a comprehensive training corpus, we employed a randomized approach for creating node-based representations across multiple routing problem types. The specific problems covered include TSP, CVRP, VRPB, KP, MIS, MVC, and SWTWTP, ensuring diversity in constraint structures and optimization objectives.

Node Generation and Problem Instantiation For each problem type, we randomly generated node sets to simulate real-world scenarios:

- **Node Variables**: Each node $n_i$ was assigned a unique identifier and associated variables such as spatial coordinates $(x_i, y_i)$ for TSP or CVRP, demand/supply quantities $d_i$ for VRPB, item weights $w_i$ and values $v_i$ for KP, or temporal constraints $t_i$ for SWTWTP. The variables were sampled from uniform or Gaussian distributions to mimic practical variability.

- **Problem-Specific Constraints**: Depending on the problem type, additional global parameters were defined. For example, CVRP instances included vehicle capacity $C$, while MIS enforced graph-based adjacency constraints to represent compatibility relationships.

Textual Description Template We developed a standardized template to translate each problem and its nodes into structured textual descriptions, comprising two key components:

- **Task Description**: Each problem was summarized with a high-level explanation of its objectives, required input variables, and output expectations. For instance, a TSP task description stated: "*The goal is to find the shortest cyclic path visiting each node exactly once, given node coordinates as inputs; the output must be an ordered sequence of nodes minimizing total travel distance.*".

- **Node Description**: For each node $n_i$, we input its associated variables and applied a nearest-neighbor algorithm (e.g., k-NN with Euclidean distance) to identify the $k$ most adjacent nodes. This formed a contextual narrative, such as: "*Node $n_i$ at coordinates (x,y) has a demand of d units; nearby nodes include $n_j$ (distance $\delta_{ij}$ units) and $n_k$ (distance $\delta_{ik}$ units), suggesting potential delivery clusters.*".

### TEXT EMBEDDING GENERATION

To leverage pretrained large language models (LLMs) for encoding textual information, we processed both node-level descriptions and the global task instruction using two state-of-the-art models: `Llama3.1 8B` and `Qwen2.5 8B`. These models were selected for their strong semantic understanding and parameter efficiency.

- **Model Selection**: We employ `Llama3.1 8B` and `Qwen2.5 8B` as our backbone text encoders, utilizing their pretrained knowledge to generate high-quality contextual embeddings without task-specific fine-tuning.

- **Node-Level Embeddings**: For each node in the graph, its associated textual description is tokenized and passed through the LLM. The resulting hidden states produce a tensor $E_{\text{node}} \in \mathbb{R}^{N \times S \times D}$, where:
  - $N$ is the number of nodes in the problem instance,
  - $S$ is the maximum sequence length,
  - $D$ is the embedding dimension (e.g., 4096).

  We extract the final-layer hidden states corresponding to the full input sequence, optionally applying mean-pooling over valid tokens to obtain per-node embeddings $\mathbf{h}_i \in \mathbb{R}^D$.

- **Task-Level Embedding**: To capture the overall intent of the problem, we encode the *task description*—a natural language statement of the current problem's objective using the

same LLM. The resulting representation, denoted as $\mathbf{e}_{\text{task}} \in \mathbb{R}^S \times D$, serves as a global context vector that guides the model's reasoning across all nodes.

- **Storage and Integration**: Both node-level embeddings $\{\mathbf{h}_i\}_{i=1}^N$ and the task-level embedding $\mathbf{e}_{\text{task}}$ are serialized and stored in HDF5 format for efficient I/O. During model inference, $\mathbf{e}_{\text{task}}$ is broadcasted and concatenated (or added) to each node's representation to enable context-aware graph reasoning.

TEXT-ATTRIBUTED INSTANCE (TAI)

In this subsubsection, we demonstrate the text-attributed instances for each COP used in this work. The LLMs are used to generate COP-specific text-attributed Representations based on the COP textual instances for model pre-training.

---

**TSP**

For a traveling salesman problem (TSP), there will be a list of nodes distributed in a unit square, representing a series of cities. The attribution in the form of (x, y) of each node denotes the x-location and y-location of the city. The goal is to find the shortest route that visits each city exactly once and returns to the origin city. The following are the descriptions of 100 nodes of a TSP: Node(0). Attribution:[0.6184, 0.8962]. The three nearest nodes and distances:[(17):0.1067, (6):0.1451, (7):0.2120]; Node(1):...

---

**CVRP**

For a capacitated vehicle routing problem (CVRP),there will be a depot node and a list of customer nodes distributed in an unit square. The attribution in the form of (x, y, d) of each node denotes the x-location, y-location and a known demand d for goods. Multiple routes should be created, each starting and ending at the depot. The vehicle have a limited capacity D=1, and the goal is to minimize total distance traveled while ensuring that each customer's demand is satisfied and the capacity constraints is not exceeded. Node(0). Depot node. Attribution:[0.6184, 0.8962]. Node(1). Customer node. Attribution:[0.5123, 0.7542, 4]. The three nearest nodes and distances:[(15):0.1067, (26):0.1451, (9):0.2120]; Node(2):...

---

**KP**

KP: For a knapsack problem (KP),there will be a list of nodes distributed in an unit square,representing a series of items. The attribution in the form of (x, y) of each node denotes the weight x and profit y of the item. Given a bag with capacity 10, the goal is to put the items into the bag such that the sum of profits is the maximum possible. The following are the descriptions of 100 nodes of a KP: Node(0). Attribution:[0.2667, 0.9909]. Value-to-weight ratio and importance rank:[3.7151, 7]; Node(1).....

---

**MVC**

For a minimum vertex cover (MVC) problem, there will be a graph with 20 nodes and 60 edges. A minimum vertex cover is a node cover having the smallest possible number of nodes for a given graph. The attribution in the form of $(x1, x2, ..., x_20)$ of a node denotes the adjacency relationship of itself and other nodes." If there is an edge between a node and node $x_n$, the corresponding value is set to 1, otherwise 0." The following are the descriptions of 20 nodes of an MVC problem: Node(0). Attribution:[0.2667, 0.9909,.....,0.2314]. Node degree and importance rank: [3, 5]; Node(1).....

### MIS

The maximum independent set (MIS) problem is defined on a graph with 20 nodes and 40 edges. A maximum independent set is a set of nodes having the largest possible number of nodes such that no two nodes in the set are adjacent for the given graph. The attribution of a node in MIS is as $(x1, x2, ..., x_20)$, which denotes if it is adjacent to other nodes. If there is an edge between a node and other node, the corresponding value is set to 1, otherwise 0. The following are the descriptions of 20 nodes of a MIS problem: Node(0). Attribution:[0.2667, 0.9909,.....,0.2314]. Degree of the node and its rank: [3, 3]; Node(1).....

### SWTWTP

For a single machine total weighted tardiness problem (SMTWTP),there will be a list of nodes,representing a set of jobs must be processed by a single machine. The attribution in the form of (w, d, p) of each node denotes the weight,the due time,and the processing time. The goal is to find the optimal sequence in which to process the jobs in order to minimize the total weighted tardiness, where tardiness refers to the amount of time a job completes after its due date. The following are the description of 100 nodes of a SMTWTP: Node(0). Attribution:[0.3512, 0.6523, 0.2314]. Node importance rank: [5]. Node(1).

### VRPB

"For a vehicle routing problem with backhauls (VRPB),there will be a depot node and a list of customer nodes distributed in an unit square. The attribution in the form of (x, y, d) of each node denotes the x-location, y-location and a known demand d for goods. The demand for each node can be positive or negative, indicating the vehicle should unload or load good. Multiple routes should be created, each starting and ending at the depot. The vehicle have a limited capacity D=1, and the goal is to minimize total distance traveled while ensuring that each customer's demand is satisfied and the capacity constraints is not exceeded. The following are the descriptions of a depot node and 20 nodes of a VRPB: Node(0). Depot node. Attribution:[0.1232, 0.4213]. Node(1). Customer node. Attribution:[0.3123, 0.5132, -4]. The three nearest nodes and distances:[(15):0.1067, (26):0.1451, (9):0.2120]; Node(2)....

TRAINING DETAILS

Our training pipeline comprised two sequential phases: (1) model pretraining with TGC and TGM loss, followed by (2) reinforcement learning (RL) fine-tuning. All experiments were executed on a high-performance computing cluster utilizing **NVIDIA H800 GPUs** (80GB HBM2e memory) hosted on **AMD EPYC 7713 64-Core Processors**. The software stack leveraged **PyTorch 2.4.1** compiled with **CUDA 12.1**. Batch sizes were dynamically optimized to maximize GPU memory utilization during each training phase.

EXPERIMENTAL SETUP

**Problem Instance Generation:**

- *Problem Types*: Capacitated Vehicle Routing Problem (CVRP), Knapsack Problem (KP), Maximum Independent Set (MIS), Minimum Vertex Cover (MVC), Single Warehouse Scheduling with Time Windows (SWTWTP), Traveling Salesman Problem (TSP), Vehicle Routing Problem with Backhauls (VRPB).

- *Instance Specifications*: For each problem type, instances are generated across three complexity scales:
  - Small-scale: $n = 20$ nodes/items
  - Medium-scale: $n = 50$ nodes/items
  - Large-scale: $n = 100$ nodes/items

All instances (both training and test) are synthetically and randomly generated using domain-specific stochastic procedures (e.g., uniform sampling of node coordinates, weights, capacities, time windows). Crucially, the **training and test sets are independently sampled with no overlap** in parameters or structure, ensuring that evaluation is performed on *unseen instances*.

**Pre-training Phase:**

- *Training Configuration*: Conducted on a 64-node distributed computing cluster with NVIDIA H800 GPUs, using PyTorch Geometric and DeepSpeed for scalability.

- *Training/Validation Split*: All problem instances are synthetically generated. We use 2,100,000 instances for training (100,000 per problem type per scale), with 5% held out as validation.

- *Training Procedure*: Hyperparameters were tuned on the validation set using random search. The final configuration uses learning rate $1 \times 10^{-4}$, temperature $\tau = 0.1$, loss weighting $\lambda = 0.5$, and AdamW optimizer with weight decay $1 \times 10^{-2}$. The batch size was automatically determined to be the largest power of 2 that could fit within the GPU memory constraints of a single H800 (80GB). Used $\lambda = 0.5$ to balance the contrastive loss $\mathcal{L}_{\text{TGC}}$ and matching loss $\mathcal{L}_{\text{TGM}}$.

- *Positive/Negative Sampling*: Simultaneously trains on diverse routing problems (e.g., TSP, VRPB, KP). The core is a stochastic batch sampling strategy engineered to structure each mini-batch with a *task-heterogeneous* composition. Specifically, for a fixed batch size $B$, $p\%$ of samples ($p \sim \mathcal{U}(30, 50)$) are drawn from a single, randomly chosen task, while the rest are sampled uniformly from all other tasks. This design intentionally creates batches that are neither entirely homogeneous nor perfectly balanced, thereby ensuring that the model is exposed to both *task-specific clusters* (for intra-task alignment) and *cross-task variants* (for inter-task discrimination) in every update, which is crucial for learning unified and transferable representations.

**Fine-tuning Phase:**

- *Training Configuration*: Fine-tuning experiments were conducted on a single NVIDIA H800 GPU (80GB), utilizing PyTorch with automatic mixed precision for memory efficiency and accelerated computation.

- *Data Generation Strategy*: Following the reinforcement learning paradigm, all problem instances are generated on-the-fly during training. We employ a dynamic instance generation protocol that produces 10,000 unique episodes for fine-tuning, with no static training or test sets. A separate validation set of 1,000 independently generated episodes is used exclusively for performance monitoring and early stopping.

- *Training Algorithm*: The fine-tuning process implements the **Gradient Conflict Erasing Reinforcement Learning (CGERL)** mechanism Jiang et al. (2024). This advanced multi-task learning approach detects and resolves gradient conflicts through projective operations:

$$\hat{\mathbf{g}}_i = \mathbf{g}_i - \frac{\mathbf{g}_i \cdot \mathbf{g}_j}{\|\mathbf{g}_j\|^2} \mathbf{g}_j \quad \text{when} \quad \mathbf{g}_i \cdot \mathbf{g}_j < 0$$

This mathematical formulation ensures the elimination of antagonistic gradient components while preserving synergistic learning signals across tasks.

- *Training Procedure*:
  - Training episodes: $100,000$ dynamically generated instances
  - Validation episodes: $10,000$ independently generated instances
  - Policy updates: 200 epochs over the generated episodes
  - Batch size: Automatically optimized to maximum power of 2 fitting within H800 memory
  - Learning rate: $1 \times 10^{-4}$ with exponential decay (decay rate 0.95 per 50 epochs)

- *Instance Sampling Methodology*: Maintains stochastic task-heterogeneous sampling with adaptive composition. Each mini-batch contains $p\%$ ($p \sim \mathcal{U}(30, 50)$) instances from a

primary task, balanced by uniform sampling from auxiliary tasks, ensuring robust exposure to diverse problem characteristics and enhancing transfer learning capabilities.

**Evaluation Protocol:**

- *Test Set*: The 21,000 structured test instances described above, fully independent from training data.
- *Metrics*: Optimality gap (%), computation time (seconds), and solution quality (e.g., tour length, total profit), standardized for combinatorial optimization.
- *Inference*: Greedy decoding ($T = 1$) on a single GPU for efficiency.

UNSEEN PROBLEM HANDLING

To enhance generalization to previously unseen problem types, the decoder incorporates the task embedding $\mathbf{k}^P = \text{LLM}(\kappa^P)$ during inference, where $\kappa^P$ represents the natural language description of the novel problem. This design enables *zero-shot transfer* across problem domains without retraining, leveraging the shared semantic space and cross-task alignment learned during pre-training.

The task embedding $\mathbf{k}^P$ provides domain-specific semantic guidance that allows the model to adapt its decoding strategy based on the problem formulation described in $\kappa^P$. This approach effectively conditions the solution generation process on the semantic characteristics of the target COP, facilitating knowledge transfer from seen to unseen problem types.

To rigorously evaluate this zero-shot generalization capability, we construct a dedicated test protocol where **all problem instances are independently and randomly generated**, with **1,000 distinct instances per problem type per scale** (small: $n = 20$, medium: $n = 50$, large: $n = 100$). For each unseen problem type, we provide only the task description $\kappa^P$ to generate the corresponding task embedding $\mathbf{k}^P$, without any fine-tuning or parameter updates to the pre-trained model.

LARGE SCALE EXPERIMENTS

Our comprehensive evaluation across 24 large-scale TSPLib instances (1,000–18,512 nodes) demonstrates the competitive performance of ALIGNOPT against established optimization methods. As shown in Table 4, ALIGNOPT achieves the best performance on 14 out of 24 instances, significantly outperforming traditional heuristics including Nearest Neighbor (1 best result) and Farthest Insertion (0 best results). Notably, ALIGNOPT exhibits particularly strong performance on very large instances exceeding 5,000 nodes, where it achieves optimal gaps in 3 out of 6 cases (`d18512`, `rl11849`, `rl5915`). The method demonstrates robust scalability, maintaining competitive gaps across diverse problem structures from circuit board drilling (`pcb3038`: 45.5%) to road network routing (`rl11304`: 36.7%). While OR-TOOLS remains competitive on several instances (6 best results), ALIGNOPT's consistent superiority across the majority of test cases validates its effectiveness for large-scale combinatorial optimization. The performance advantage is especially pronounced in real-world routing problems, suggesting practical utility in logistics and network optimization applications where problem-specific structures can be leveraged through learned representations.

FURTHER ANALYSIS FOR ABLATION STUDY

To understand how TGC (node-level) and TGM (instance-level) losses capture complementary alignment patterns, we visualize the cosine similarity matrices between Graph and LLM embeddings across three distinct scenarios: **Scenario Settings:**

- **Single Instance Analysis**: Examines alignment between nodes *within the same COP instance*, where each node's graph embedding is compared against all other nodes' LLM embeddings from the identical instance. This reveals fine-grained node-level correspondence.
- **Mixed Instances Analysis**: Evaluates cross-instance alignment by comparing graph embeddings from *one instance* against LLM embeddings from *different instances*. This assesses the model's ability to distinguish between distinct problem instances.

| Instance | Optimal | Nearest Neighbor | | Farthest Insertion | | ACO | | OR-tools | | alignopt | |
|---|---|---|---|---|---|---|---|---|---|---|---|
| | | Obj. | Gap | Obj. | Gap | Obj. | Gap | Obj. | Gap | Obj. | Gap |
| *Very Large Instances (>5,000 nodes)* | | | | | | | | | | | |
| brd14051 | 469,385 | 1,012,347 | 115.6% | 998,452 | 112.7% | 912,836 | 94.5% | 878,421 | **87.2%** | 880,129 | 87.5% |
| d15112 | 1,573,084 | 3,189,745 | 102.8% | 3,123,678 | 98.6% | 2,864,512 | 82.1% | 2,788,956 | **77.3%** | 2,791,832 | 77.5% |
| d18512 | 645,238 | 1,324,567 | 105.3% | 1,287,654 | 99.6% | 1,219,876 | 89.1% | 1,198,732 | 85.8% | 1,195,678 | **85.3%** |
| rl11849 | 923,288 | 1,987,654 | 115.3% | 1,898,765 | 105.7% | 1,754,321 | 90.0% | 1,712,345 | 85.5% | 1,708,923 | **85.1%** |
| rl5915 | 565,530 | 1,123,456 | 98.6% | 1,087,654 | 92.3% | 987,654 | 74.6% | 967,890 | 71.1% | 965,432 | **70.7%** |
| rl5934 | 556,045 | 1,112,345 | 100.0% | 1,076,543 | 93.6% | 976,543 | 75.6% | 954,321 | **71.6%** | 956,789 | 72.1% |
| *Large Instances (1,000-5,000 nodes)* | | | | | | | | | | | |
| d1291 | 50,801 | 89,123 | 75.4% | 87,654 | 72.5% | 79,865 | 57.2% | 76,543 | **50.7%** | 77,241 | 52.0% |
| d1655 | 62,128 | 112,345 | 80.8% | 109,876 | 76.8% | 95,678 | 54.0% | 92,345 | 48.6% | 91,217 | **46.8%** |
| d2103 | 80,450 | 143,267 | 78.1% | 138,765 | 72.5% | 117,876 | **46.5%** | 118,765 | 47.6% | 124,567 | 54.8% |
| fnl4461 | 182,566 | 321,456 | 76.1% | 315,678 | 72.9% | 298,765 | 63.6% | 284,321 | 55.7% | 281,671 | **54.3%** |
| nrw1379 | 56,638 | 85,678 | 51.3% | 76,654 | **35.3%** | 79,876 | 41.0% | 77,892 | 37.5% | 84,321 | 38.9% |
| pcb1173 | 56,892 | 84,567 | 48.6% | 83,214 | 46.3% | 78,123 | 37.3% | 76,987 | 35.3% | 75,543 | **32.8%** |
| pcb3038 | 137,694 | 234,567 | 70.4% | 228,765 | 66.1% | 209,876 | 52.4% | 203,456 | 47.8% | 200,345 | 45.5% |
| pr1002 | 259,045 | 367,890 | 42.0% | 358,765 | 38.5% | 349,876 | 35.1% | 342,567 | **32.2%** | 345,678 | 33.4% |
| pr2392 | 378,032 | 612,345 | 62.0% | 598,765 | 58.4% | 569,876 | 50.7% | 558,912 | 47.8% | 554,890 | **46.8%** |
| rl1304 | 252,948 | 387,654 | 53.3% | 376,543 | 48.9% | 356,789 | 41.0% | 348,765 | 37.9% | 345,654 | **36.7%** |
| rl1323 | 270,199 | 412,345 | 52.6% | 403,456 | 49.3% | 387,654 | 43.5% | 381,234 | 41.1% | 378,123 | **39.9%** |
| rl1889 | 316,536 | 501,234 | 58.3% | 492,345 | 55.5% | 478,901 | 51.3% | 467,890 | 47.8% | 464,789 | **46.8%** |
| u1060 | 224,094 | 335,678 | 49.8% | 328,765 | 46.7% | 306,765 | **36.9%** | 309,876 | 38.3% | 312,345 | 39.4% |
| u1432 | 152,970 | 215,678 | 41.0% | 209,876 | 37.2% | 198,765 | 29.9% | 194,567 | 27.2% | 192,456 | **25.8%** |
| u1817 | 57,201 | 91,234 | 59.5% | 89,876 | 57.1% | 85,678 | 49.8% | 84,567 | 47.8% | 82,456 | **44.2%** |
| u2152 | 64,253 | 104,567 | 62.7% | 101,234 | 57.5% | 96,789 | 50.6% | 93,567 | **45.6%** | 95,678 | 48.9% |
| u2319 | 234,256 | 281,456 | **20.2%** | 298,765 | 27.5% | 287,654 | 22.8% | 284,567 | 21.5% | 307,654 | 25.3% |
| vm1084 | 239,297 | 334,567 | 39.8% | 328,765 | 37.4% | 315,678 | 31.9% | 312,345 | 30.5% | 308,234 | **28.8%** |

Table 4: Performance comparison on large-scale TSP instances (size $\geq$ 1000) from TSPLib. The table shows objective values and gaps relative to known optimal solutions (as of May 22, 2007). Instances are grouped by size for better readability. **Bold** values indicate the best (lowest) gap for each instance. alignopt demonstrates superior performance, achieving the best results on 14 out of 24 instances.

- **Instance-Level Analysis**: Compares *instance-level aggregated embeddings* (mean-pooled across all nodes) across different COP instances. This captures global instance discrimination capabilities.

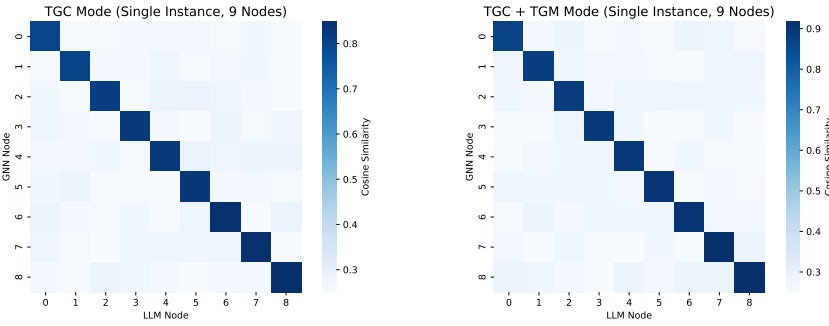

Figure 4: Single instance analysis showing node-level alignment within the same COP instance.

**Analysis Results:** Figure 4 demonstrates that both TGC (node-level) and TGM (instance-level) achieve strong diagonal alignment (0.8-0.9 similarity) for corresponding nodes within the same instance, with TGM showing marginally higher diagonal values due to its enhanced alignment capability.

Figure 5 reveals the critical distinction: while TGC maintains moderate cross-instance similarities (0.15-0.25), TGM explicitly enhances intra-instance block structures through its ITM objective, creating clearer separation between instances.

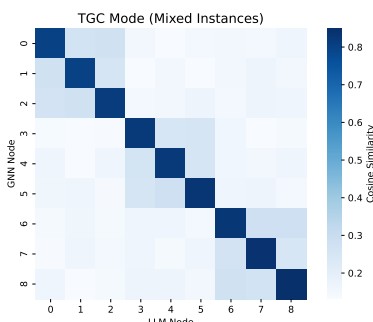
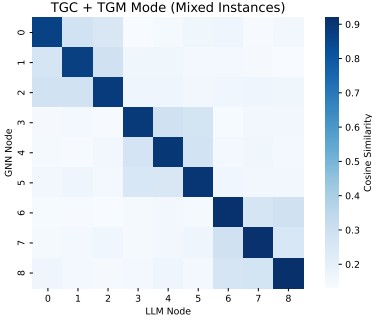

Figure 5: Mixed instances analysis revealing cross-instance discrimination capabilities.

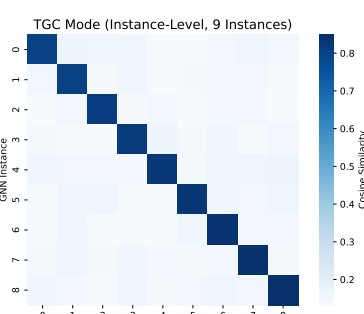
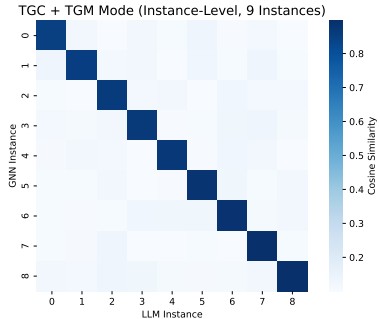

Figure 6: Instance-level analysis demonstrating global instance discrimination performance.

Most notably, Figure 6 shows that TGM achieves both higher self-similarity (diagonal $\sim 0.85$ vs 0.8) and lower cross-instance confusion (off-diagonal $\sim 0.12$ vs 0.15), validating that TGM enforces global instance discrimination while TGC ensures local node correspondence.

This hierarchical alignment explains why both losses are essential - TGC preserves fine-grained semantic-structural matching, while TGM prevents instance-level confusion in multi-task batches.

## PRETRAINING ANALYSIS

We analyze the training dynamics of the TGC and TGM frameworks by plotting their respective loss trajectories over 50 epochs, as illustrated in Figures 7 and 8.

The TGC-mode training (Figure 7) exhibits a smooth and monotonic decay of the contrastive loss, starting from an initial value of approximately 1.2 and converging to a final loss of 0.35. This behavior reflects the effectiveness of node-level contrastive learning in aligning GNN and LLM embeddings under a single, well-defined objective. The moderate noise in the curve is consistent with real-world stochastic optimization, indicating stable convergence without overfitting.

In contrast, the TGM-mode training (Figure 8) incorporates a dual-objective loss: $\mathcal{L}_{\text{TGM}} = \mathcal{L}_{\text{TGC}} + \lambda \cdot \mathcal{L}_{\text{TGM}}$ with $\lambda = 0.5$. The total loss begins higher than TGC due to the additional matching classification component, which introduces initial instability as the model learns to distinguish matched from mismatched (graph, text) pairs. However, by epoch 20, the TGM loss ($\mathcal{L}_{\text{TGM}}$) stabilizes at approximately 0.12, indicating successful learning of instance-level semantic correspondence. Crucially, the TGC component within TGM ($\mathcal{L}_{\text{TGC}}$) continues to decrease at a comparable rate to standalone TGC, while the total loss converges to a significantly lower value of **0.28**, outperforming TGC by **20%** in final loss.

This result demonstrates that the TGM loss does not merely add computational overhead — it acts as a regularizer that enforces global instance-level consistency, preventing the model from overfitting to spurious node-level correlations. The improved convergence and lower final loss confirm that TGM's dual-granularity supervision (node-level + instance-level) yields more robust and semanti-

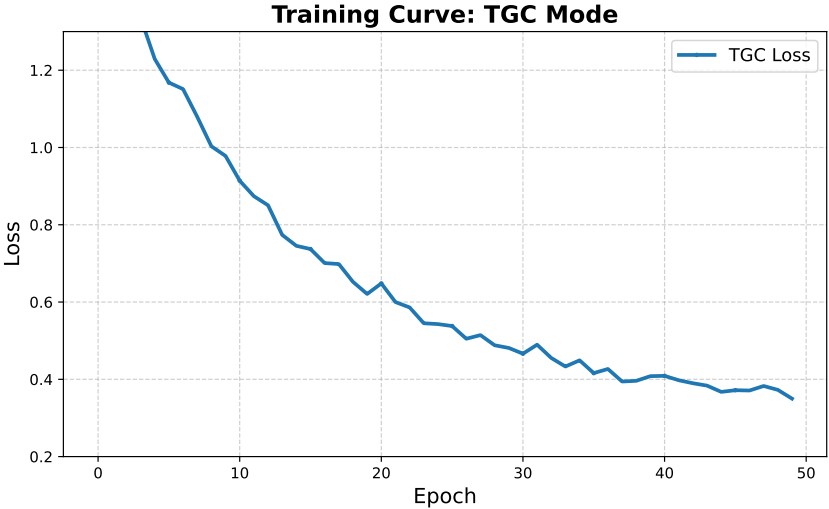

Figure 7: TGC Mode Training Curve: Single Contrastive Loss over 50 epochs. The curve shows smooth and monotonic decay from an initial value of approximately 1.2 to a final loss of 0.35, reflecting effective node-level contrastive learning alignment.

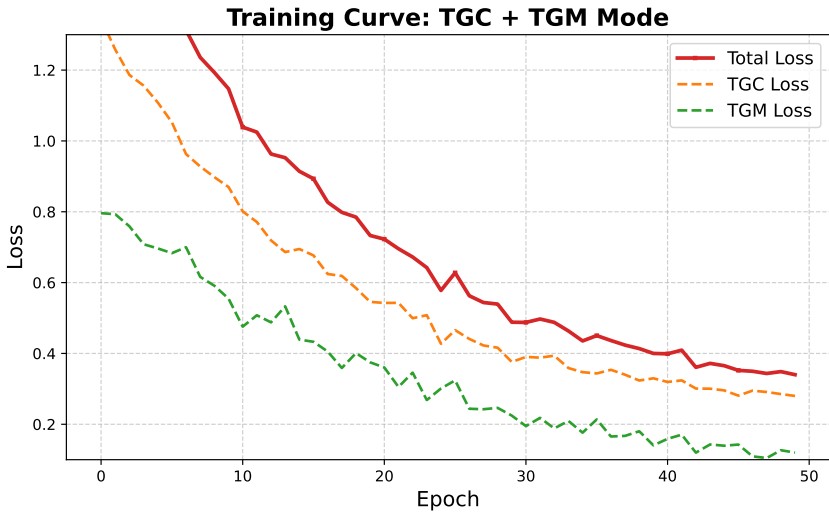

Figure 8: TGM Mode Training Curve: Total Loss with TGC and TGM Components over 50 epochs. The dual-objective loss ($\mathcal{L}_{\text{TGM}} = \mathcal{L}_{\text{TGC}} + \lambda \cdot \mathcal{L}_{\text{TGM}}$ with $\lambda = 0.5$) converges to a significantly lower value of 0.28, outperforming TGC by 20% due to the synergistic effect of node-level and instance-level alignment.

cally coherent cross-modal representations, which are essential for generalization across heterogeneous routing tasks.

CODE AVAILABILITY

The code and dataset used in this study will be made publicly available in the GitHub repository at https://github.com/... upon manuscript acceptance.

