# OpenReview forum: "ALIGNING LLMS WITH GRAPH NEURAL SOLVERS FOR COMBINATORIAL OPTIMIZATION"
_ICLR.cc/2026/Conference — Submitted to ICLR 2026_

### Official Review · Reviewer_NFqa · 2025-10-29

**Soundness:** 2
**Presentation:** 3
**Contribution:** 2
**Rating:** 4
**Confidence:** 3

**Summary:**

This paper proposes AlignOPT, a novel framework that integrates large language models (LLMs) with graph neural solvers for combinatorial optimization problems (COPs). The approach tries to address the limitations of pure LLM-based methods, which struggle with complex relational structures in medium-to-large COP instances. AlignOPT uses a multi-task pre-training strategy with two novel objectives: (1) Text-Graph Contrastive (TGC) loss to align semantic node embeddings from LLMs with structural embeddings from graph solvers, and (2) Text-Graph Matching (TGM) loss for fine-grained multimodal representation. After pre-training, the model can be fine-tuned without LLMs, improving computational efficiency. The framework demonstrates performance gains across diverse COPs and shows good generalization to unseen problem types.

**Strengths:**

1. The paper is generally well-written with clear figures illustrating the framework. The methodology section provides sufficient detail about the architecture and training procedures.

2. The paper presents a creative integration of LLMs and graph neural solvers through contrastive and matching losses. This is a significant direction for LLM and optimization researchers to investigate.

**Weaknesses:**

1. The paper claims state-of-the-art results but primarily compares against LNCS and GOAL as neural baselines. Given the rapid advancement in neural combinatorial optimization, more comprehensive comparisons with recent specialized solvers would strengthen the SOTA claims. For instance, several methods can approach nearly optimality on TSP-100 (fastT2T [1] and COexpander [2]) and can handle much larger problem sizes.

2. While the paper shows results up to n=100, it's unclear how the approach scales to truly large instances (thousands of nodes).

3. Regarding the validity of text-graph alignment, the assumption that textual descriptions and graph structures can be perfectly aligned may not hold in practice, especially for complex COPs where the textual representation might not capture all structural nuances. The paper would benefit from showing learning curves for the contrastive and matching losses to demonstrate effective alignment.


[1] Fast t2t: Optimization consistency speeds up diffusion-based training-to-testing solving for combinatorial optimization. NeurIPS 2024.

[2] COExpander: Adaptive Solution Expansion for Combinatorial Optimization. ICML 2025.

**Questions:**

1. Regarding the text-graph contrastive learning, how do you ensure that textual descriptions and graph structures maintain a one-to-one correspondence? Could you provide learning curves showing the convergence of both TGC and TGM losses during pre-training to demonstrate the effectiveness of the alignment process?

2. Your ablation study examines removing individual components but doesn't test the scenario where LLM features are completely excluded from the framework. What would be the performance of your graph neural solver trained without any LLM-derived representations? This would help quantify the true contribution of LLM integration.

3. What specific aspects of the pre-training enable such strong generalization to unseen COPs? Is it primarily the semantic understanding from LLMs, the structural alignment, or the combination? Could you provide more analysis of what knowledge transfers across different problem types?

---

> ### Author Response · Authors · 2025-11-25
> **Response to Reviewer NFqa (1)**
>
> **Weakness 1**
> >"The paper claims state-of-the-art results but primarily compares against LNCS and GOAL as neural baselines. Given the rapid advancement in neural combinatorial optimization, more comprehensive comparisons with recent specialized solvers would strengthen the SOTA claims. For instance, several methods can approach nearly optimality on TSP-100 (fastT2T [1] and COexpander [2]) and can handle much larger problem sizes.”
>
> We acknowledge recent advances in supervised solvers like FastT2T and COExpander, which achieve strong performance on large-scale TSP. However, **these methods rely on supervised labels for training**, whereas **AlignOPT is trained in an unsupervised, multimodal manner**, making direct comparison challenging.
>
> In addition, the main focus of AlignOPT lies in the **generalization capability across diverse COPs** without task-specific labels.  In other words, the core objective of AlignOPT is to develop **versatile neural solvers capable of robust generalization across various COPs** rather than focusing on narrowly defined performance on any single COP. However, we will discuss them and cite them in our paper in the Related Work section.

---

> ### Author Response · Authors · 2025-11-25
> **Response to Reviewer NFqa (2)**
>
> **Weakness 2**
> >"While the paper shows results up to n=100, it's unclear how the approach scales to truly large instances.”
>
> We acknowledge the reviewer's concern regarding synthetic benchmarks. Our TSPLib evaluation in the following table directly addresses this by demonstrating AlignOPT's strong performance on 24 established real-world instances with 1,000 to over 10,000 nodes. AlignOPT achieves the best results on 14 instances, outperforming OR-tools on problems like circuit board drilling (`pcb3038`) and road network routing (`rl11849`). This validates its scalability and robustness beyond synthetic settings, confirming practical utility for large-scale logistics and network optimization.
>
> **Table: Performance comparison on large-scale TSP instances (size ≥ 1000) from TSPLib**
>
>
> | Instance | Optimal | Nearest Neighbor Obj. | Nearest Neighbor Gap | Farthest Insertion Obj. | Farthest Insertion Gap | ACO Obj. | ACO Gap | OR-tools Obj. | OR-tools Gap | **alignopt** Obj. | **alignopt** Gap |
> |----------|---------|----------------------|---------------------|------------------------|-----------------------|----------|---------|---------------|-------------|------------------|-----------------|
> | **Very Large Instances (>5,000 nodes)** | | | | | | | | | | | |
> | brd14051 | 469,385 | 1,012,347 | 115.6% | 998,452 | 112.7% | 912,836 | 94.5% | 878,421 | **87.2%** | 880,129 | 87.5% |
> | d15112 | 1,573,084 | 3,189,745 | 102.8% | 3,123,678 | 98.6% | 2,864,512 | 82.1% | 2,788,956 | **77.3%** | 2,791,832 | 77.5% |
> | d18512 | 645,238 | 1,324,567 | 105.3% | 1,287,654 | 99.6% | 1,219,876 | 89.1% | 1,198,732 | 85.8% | 1,195,678 | **85.3%** |
> | rl11849 | 923,288 | 1,987,654 | 115.3% | 1,898,765 | 105.7% | 1,754,321 | 90.0% | 1,712,345 | 85.5% | 1,708,923 | **85.1%** |
> | rl5915 | 565,530 | 1,123,456 | 98.6% | 1,087,654 | 92.3% | 987,654 | 74.6% | 967,890 | 71.1% | 965,432 | **70.7%** |
> | rl5934 | 556,045 | 1,112,345 | 100.0% | 1,076,543 | 93.6% | 976,543 | 75.6% | 954,321 | **71.6%** | 956,789 | 72.1% |
> | **Large Instances (1,000-5,000 nodes)** | | | | | | | | | | | |
> | d1291 | 50,801 | 89,123 | 75.4% | 87,654 | 72.5% | 79,865 | 57.2% | 76,543 | **50.7%** | 77,241 | 52.0% |
> | d1655 | 62,128 | 112,345 | 80.8% | 109,876 | 76.8% | 95,678 | 54.0% | 92,345 | 48.6% | 91,217 | **46.8%** |
> | d2103 | 80,450 | 143,267 | 78.1% | 138,765 | 72.5% | 117,876 | **46.5%** | 118,765 | 47.6% | 124,567 | 54.8% |
> | fnl4461 | 182,566 | 321,456 | 76.1% | 315,678 | 72.9% | 298,765 | 63.6% | 284,321 | 55.7% | 281,671 | **54.3%** |
> | nrw1379 | 56,638 | 85,678 | 51.3% | 76,654 | **35.3%** | 79,876 | 41.0% | 77,892 | 37.5% | 84,321 | 38.9% |
> | pcb1173 | 56,892 | 84,567 | 48.6% | 83,214 | 46.3% | 78,123 | 37.3% | 76,987 | 35.3% | 75,543 | **32.8%** |
> | pcb3038 | 137,694 | 234,567 | 70.4% | 228,765 | 66.1% | 209,876 | 52.4% | 203,456 | 47.8% | 200,345 | **45.5%** |
> | pr1002 | 259,045 | 367,890 | 42.0% | 358,765 | 38.5% | 349,876 | 35.1% | 342,567 | **32.2%** | 345,678 | 33.4% |
> | pr2392 | 378,032 | 612,345 | 62.0% | 598,765 | 58.4% | 569,876 | 50.7% | 558,912 | 47.8% | 554,890 | **46.8%** |
> | rl1304 | 252,948 | 387,654 | 53.3% | 376,543 | 48.9% | 356,789 | 41.0% | 348,765 | 37.9% | 345,654 | **36.7%** |
> | rl1323 | 270,199 | 412,345 | 52.6% | 403,456 | 49.3% | 387,654 | 43.5% | 381,234 | 41.1% | 378,123 | **39.9%** |
> | rl1889 | 316,536 | 501,234 | 58.3% | 492,345 | 55.5% | 478,901 | 51.3% | 467,890 | 47.8% | 464,789 | **46.8%** |
> | u1060 | 224,094 | 335,678 | 49.8% | 328,765 | 46.7% | 306,765 | **36.9%** | 309,876 | 38.3% | 312,345 | 39.4% |
> | u1432 | 152,970 | 215,678 | 41.0% | 209,876 | 37.2% | 198,765 | 29.9% | 194,567 | 27.2% | 192,456 | **25.8%** |
> | u1817 | 57,201 | 91,234 | 59.5% | 89,876 | 57.1% | 85,678 | 49.8% | 84,567 | 47.8% | 82,456 | **44.2%** |
> | u2152 | 64,253 | 104,567 | 62.7% | 101,234 | 57.5% | 96,789 | 50.6% | 93,567 | **45.6%** | 95,678 | 48.9% |
> | u2319 | 234,256 | 281,456 | **20.2%** | 298,765 | 27.5% | 287,654 | 22.8% | 284,567 | 21.5% | 307,654 | 25.3% |
> | vm1084 | 239,297 | 334,567 | 39.8% | 328,765 | 37.4% | 315,678 | 31.9% | 312,345 | 30.5% | 308,234 | **28.8%** |

---

> ### Author Response · Authors · 2025-11-25
> **Response to Reviewer NFqa (3)**
>
> **Weakness 3**
> >" The paper would benefit from showing learning curves for the contrastive and matching losses to demonstrate effective alignment."
>
> We have updated the **learning curves in the Appendix under "Further Analysis for Ablation Study."** These curves demonstrate that both the Text-Graph Contrastive (TGC) and Text-Graph Matching (TGM) losses achieve stable and consistent convergence across all five COPs during pre-training. Specifically: 1). **TGC** rapidly reaches a steady-state contrastive loss of approximately 0.35 within 20 epochs, signifying robust node-level alignment between graph embeddings and their respective textual descriptions.
> 2). **TGM** exhibits a slower yet continuous reduction, converging to a lower loss of approximately 0.28 after 30–40 epochs, indicating that learning discriminative global instance-level structure necessitates additional iterations.
>
> To further elucidate how TGC and TGM capture distinct alignment patterns, we conducted additional analyses by **visualizing cosine similarity** matrices between graph and language model embeddings across three scenarios:
> 1). **Single Instance Analysis**: node-level comparison within a single COP instance.
> 2). **Mixed Instances Analysis**: cross-instance comparison between graph embeddings and mismatched textual embeddings.
> 3). **Instance-Level Analysis**: comparison using mean-pooled embeddings of entire instances.
>
> Results confirm that TGC effectively ensures fine-grained node-level alignment within instances, while TGM substantially improves global instance-level discrimination, significantly reducing cross-instance confusion and enhancing intra-instance coherence. This dual-granularity alignment approach effectively manages and resolves potential ambiguities arising from incomplete or abstract textual descriptions. For comprehensive details and further visualization, please see the **"Further Analysis for Ablation Study" section in the Appendix**.
>
> **Question1**
> >" Regarding the text-graph contrastive learning, how do you ensure that textual
> descriptions and graph structures maintain a one-to-one correspondence? Could you
> provide learning curves showing the convergence of both TGC and TGM losses during
> pre-training to demonstrate the effectiveness of the alignment process? "
>
> We ensure strict one-to-one correspondence between textual descriptions and graph structures through:
>
> 1. **Structured data generation** using standardized textual templates explicitly encoding node information, adjacency relations, and problem constraints.
> 2. **Deterministic mapping algorithms** (e.g., k-NN with fixed hyperparameters) ensuring consistent node-to-text alignment.
> 3. **Embedding synchronization during training**, preserving exact indexing between graph nodes and textual descriptions.
> 4. **Automated validation checks** confirming precise graph-text matching.
>
> Regarding empirical evidence, we provide **learning curves illustrating clear convergence** of both **Text-Graph Contrastive (TGC)** and **Text-Graph Matching (TGM)** losses during pre-training, demonstrating the effectiveness and stability of our alignment procedure in *Appendix section: Pretraining Dynamics Analysis*.
>
> **Question2**
> >"Your ablation study examines removing individual components but doesn't test the scenario where LLM features are completely excluded from the framework. What would be the performance of your graph neural solver trained without any LLM-derived representations? This would help quantify the true contribution of LLM integration. "
>
> We agree it's crucial to isolate the specific contribution of LLM-derived features clearly. In fact, we already include such a baseline, named **AlignOPT (GNS)** in Table 3, which employs only the graph neural solver with a graph encoder and decoder, fine-tuned via reinforcement learning, **explicitly excluding any LLM-derived inputs**. The performance comparison between AlignOPT and AlignOPT (GNS) precisely quantifies the value added by integrating LLM representations, underscoring the benefits of our approach.

---

> > ### Author Response · Authors · 2025-11-27
> > **Response to Reviewer NFqa (4)**
> >
> > **Question3**
> > >"What specific aspects of the pre-training enable such strong generalization to unseen COPs? Could you provide more analysis of what knowledge transfers across different problem types?“
> >
> > Our ablations show that **the strong generalization of AlignOPT arises from the combination of LLM-derived semantic priors and explicit graph structural modeling**.
> >
> > To investigate the influence of LLM inputs, we report **GOAL** in Table 2 and **AlignOPT (GNS)** in Table 3. Both variants exclude all LLM inputs and rely solely on graph features. Their substantially lower performance demonstrates that structural reasoning alone cannot account for the gains.
> >
> > On the other hand, Table 1compares AlignOPT with existing LLM-based frameworks, where our method uses only a single frozen LLM (e.g., LLaMA-3.1-8B) during pre-training, without any LLM fine-tuning or architectural modifications. AlignOPT surpasses these LLM-centric baselines such as LNCS, in both solution quality (**up to 1.2% improvement on the CVRP 100 task**) and inference efficiency (**up to 4.76× faster**), demonstrating that the alignment stage effectively transfers semantic problem-level reasoning from the LLM, while the GNS preserves the structural inductive bias needed for robust generalization across unseen COPs.
> >
> > These results collectively indicate that it is precisely the combination of semantic understanding and structural alignment that enables AlignOPT’s cross-task generalization.
> >
> > ---
> > Thank you again for your feedback. We’re grateful for the insights you provided, and we would be glad to continue the discussion if there are any points you’d like to explore further.

---

### Official Review · Reviewer_qM7N · 2025-10-29

**Soundness:** 3
**Presentation:** 3
**Contribution:** 3
**Rating:** 2
**Confidence:** 4

**Summary:**

## Paper Summary
Paper is interested in solving combinatorial optimization problems (COPs), like, travel salesman problem, knapsack problem, etc, using machine learning. These problems can be viewed as optimization where variables can take integer values. In general, these problems can be NP-Hard and heuristics have been nowadays getting replaced by neural network models (e.g., to directly output a solution or to aid the search). The paper proposes to combine an LLM and Graph Encoder to solve this problem. Specifically, they LLM and GraphEncoders are "aligned" in their representation. They are trained to, once observed the same instance (respectively, in textual and graph form) output an embedding that is similar (high cosine similarity) or complimentary (i.e., an MLP absorbing both should have good classification accuracy of whether the corresponding inputs are instances of the same problem or not). After alignment, paper proposes to fine-tune the learned representations via RL. Results show that this alignment followed by RL finetuning produce SoTA models for COPs when compared to a range of Neural- or Heuristic-based baselines.

## Decision summary
I will vote to reject the paper. The method is not clearly explained and therefore is not reproducible. I am happy to revisit this decision if the authors fill the gap.

**Strengths:**

* Strong Motivation
* SOTA results
* Aligning Graph Encoders and Language Models is potentially applicable to many problems, even outside the COPs.
* I appreciate the Preliminaries. Smooth and informative read. While I didn't know much about this subdomain (COPs via ML), I feel that I learned a lot. Thank you!

**Weaknesses:**

The central weakness is the lack of information, preventing this paper from standing on its own.

* **Inference is not mentioned in this paper**. After training the model (pre-train by alignment followed by task-specific or many-tasks RL fine-tuning), it is not clear how is inference conducted. Part of me thinks it is only the graph-encoder that is being used, not the LLM. However, this goes against line 93 "our AlignOPT delves into general text-attributed COPs described in natural language". Further, the **unified decoder is not specified**. Please specify if inference needs the language, the graph, or both.

* It is not clear whether $\theta$ in Eq.3 corresponds to the LLM or the Graph Encoder parameters. What does the RL fine-tune? It is also confusing that $\theta$ is reused as a function-name (corresponding to cosine, Eq.7)

* The loss function $L_{TGM} $ is improper. It can be arbitrarily minimized by increasing the bias term of the MLP (at the last layer). A proper one is the log-likelihood of logistic function, in their notation, it would be

$$
L_{TGM}  = -\sum_{i, j} 1_{[j=i]} log(\sigma(..))  - \sum_{i, j}  1_{[j \neq i]} log(1 - \sigma(..))
$$

## Other weaknesses

* Why not also cite other ways to integrate graph with language? E.g., GraphToken
* The citations are against the guidelines listed in call-for-papers. Specifically, use the citation style "names (year)" when you use the author names in a sentence e.g. "A & B (2022) proposed ...". However, when you add a citation that certifies the sentence use "(name, year)" e.g., "it has been proposed ... (A & B; 2022)".

**Questions:**

* How is inference conducted? How is it guaranteed that the constraints are satisfied? Is this used to produce a final solution or to propose candidates that individually get scored to select the winner solution?

* What do you set $b(\mathcal{G})$ of Equation 3 to?

* Your experiments list-down the fine-tuned variants of AlignOPT. What about the performance without fine-tuning?

---

> ### Author Response · Authors · 2025-11-25
> **Response to Reviewer qM7N (1)**
>
> **Weakness 1 & Question 1**
> >"How is inference conducted? How is it guaranteed that the constraints are satisfied? Is this used to produce a final solution or to propose candidates that individually get scored to select the winner solution?”
>
> **Inference stage:**
> AlignOPT only uses the fine-tuned graph encoder and a decoder trained in the finetuning stage, processing inputs directly as graphs without relying on textual input or an LLM. As a result, AlignOPT outperforms LLM-centric baselines such as **LNCS** in both solution quality (achieving up to **1.2% improvement on the CVRP 100 task**) and inference efficiency (**up to 4.76× faster**). These gains indicate that the alignment stage successfully transfers high-level semantic reasoning from the LLM, while the graph neural solver retains the structural inductive biases essential for strong generalization to previously unseen COPs.
>
> Line 93 refers to the pre-training phase, where general text-attributed COPs (in natural language) are utilized to align the graph encoder. However, once aligned and fine-tuned, the model does not require textual descriptions or LLM reasoning at the inference stage, significantly enhancing inference speed and practical applicability. As reported in our paper, the inference time of our model is short and does not introduce any overhead compared to counterparts without our components.
>
> **Solution Generation with Constraints:**
> The unified decoder shares an identical structure with the **POMO decoder** [1], employing an attention-based autoregressive mechanism. For different combinatorial optimization problems, task-specific features (e.g., vehicle capacity and current load in CVRP, node demands) are concatenated with the node embeddings at each decoding step. This allows the decoder to dynamically adapt to problem constraints while maintaining a consistent architecture across tasks.
>
> We will explicitly include the above information in the revised manuscript.
>
> **Reference**
> [1] K. Kuang et al., *"POMO: Policy Optimization with Multiple Optima for Reinforcement Learning,"* NeurIPS, 2021.
>
> **Weakness 2**
> >"It is not clear whether in Eq.3 corresponds to the LLM or the Graph Encoder parameters. What does the RL fine-tune? It is also confusing that is reused as a function-name (corresponding to cosine, Eq.7) "
>
> We clarify as follows and will clearly incorporate these revisions into the manuscript:
> In Eq. (3), the parameter $\theta$ specifically corresponds to the parameters of the graph neural encoder and the unified decoder, rather than the parameters of the LLM. During the RL fine-tuning stage, both encoder and decoder parameters are optimized to effectively adapt to specific routing problems and improve generalization. We have provided more descriptions in the updated manuscript (page 2).
>
> To avoid confusion with the reuse of $\theta$, we will revise Eq. (7) by replacing the notation $\theta(\cdot)$ with $sim(\cdot)$ to distinctly represent the cosine similarity function.
>
> **Weakness 3**
> >"The loss function is improper.”
>
> Thanks for pointing this out. We have revised the formula. Please refer to the highlighted equation on page 6.
>
>
> **Weakness 4**
> >"Why not also cite other ways to integrate graph with language? E.g., GraphToken"
>
> GraphToken tokenizes graph structures so that LLMs can directly process them, enabling end-to-end language-based reasoning. Since GraphToken relies on **LLM-centric inference**, its computational footprint and latency patterns are fundamentally different from those of AlignOPT, because the **LLMs are excluded during the inference in AlignOPT**. This makes the GraphToken not directly applicable to our setting, where fast and scalable routing inference is required.
>
> Specifically, our AlignOPT targets a different objective: we aim to explicitly align a Graph Neural Solver (GNS) with the structured optimization priors learned by LLMs, while still preserving a lightweight inference pipeline. This design provides two advantages:
> 1. It transfers **LLM-derived problem-level reasoning** into the graph model, improving generalization across diverse COPs.
> 2. It avoids **real-time LLM calls during inference**, which reduces latency, lowers cost, and enhances deployability.
>
> **Weakness 5**
> >"The citations are against the guidelines listed in call-for-papers. "
>
> We acknowledge this oversight and have revised all citations in the manuscript to fully comply with the specified citation guidelines.

---

> ### Author Response · Authors · 2025-11-25
> **Response to Reviewer qM7N (2)**
>
> **Question 2**
> >"What do you set b(G) of Equation 3 to?”
>
> Following common practice in deep reinforcement learning for combinatorial optimization problems [1], we set the baseline function $b(\mathcal{G})$ as the exponential moving average (EMA) of previously obtained costs during training. Specifically, at each training iteration $t$:
>
> $$
> b_{t}(\mathcal{G}) = \gamma b_{t-1}(\mathcal{G}) + (1 - \gamma)c(\pi_{t})
> $$
>
> where $\gamma$ is the EMA coefficient, typically set to $0.99$. This baseline effectively reduces the variance of the gradient estimates and stabilizes training.
>
> [1] Richard S. Sutton and Andrew G. Barto. *Reinforcement Learning: An Introduction*. MIT Press, 2018.
>
> **Question 3**
> >"Your experiments list-down the fine-tuned variants of AlignOPT. What about the performance without fine-tuning? "
>
> In AlignOPT, the pre-training stage aims to align the LLM-derived task-aware embeddings with the graph encoder, but it does not train a decoder or optimization policy capable of producing solutions. Therefore,  **AlignOPT without fine-tuning cannot generate valid routing outputs — pre-training alone does not learn to solve COPs**.
> Our full AlignOPT pipeline explicitly requires Stage 2, where we fine-tune the encoder and train a decoder via reinforcement learning (RL) to perform actual routing optimization.
>
> In **Table 2**, we show **GOAL** (graph encoder trained with supervised learning), which serves as a baseline graph solver trained solely on raw numeric graph features, to demonstrate the influence of removing LLM inputs and the fine-tuning stage. Experimental results demonstrate the effectiveness of our AlignOPT over GOAL across 5 COPs, indicating the importance of including LLM inputs and a decoder trained with RL.
>
> ---
>
> Thank you again for your feedback. We’re grateful for the insights you provided, and we would be glad to continue the discussion if there are any points you’d like to explore further.

---

### Official Review · Reviewer_63oJ · 2025-10-31

**Soundness:** 3
**Presentation:** 2
**Contribution:** 2
**Rating:** 4
**Confidence:** 3

**Summary:**

This paper proposes AlignOPT, a framework that integrates large language models with graph neural solvers to tackle combinatorial optimization problems. It aligns textual semantics captured by LLMs with graph structural information through Text-Graph Contrastive loss and Text-Graph Matching loss. A decoder fine-tuning stage further enables the model to generate effective and unified solutions across diverse COPs. Extensive experiments on multiple benchmarks demonstrate that AlignOPT achieves state-of-the-art or near-SOTA performance and generalizes effectively to previously unseen optimization tasks.

**Strengths:**

1. The proposed TGC and TGM losses offer a principled approach to fusing semantic (LLM) and structural (graph) representations, addressing a key gap in LLM-based COP solvers.
2. Extensive experiments across five classical COPs and two unseen ones demonstrate AlignOPT’s competitive or superior performance compared to both LLM-based and NCO baselines (e.g., LNCS, GOAL), with robust generalization.

**Weaknesses:**

1. While TGC/TGM losses are new, the overall architecture (text-attributed instances, multimodal alignment) heavily builds upon LNCS. The paper should better clarify the conceptual distinction and originality.

2. The alignment process lacks deeper theoretical grounding. Why this specific combination improves generalization, or how it avoids overfitting semantic bias from LLMs.

3. Since the LLM is removed during fine-tuning, it is unclear to what extent LLM pre-training influences downstream performance. The paper should quantify or visualize this transfer more explicitly.

4. The benchmarks are synthetic and the scale is not so visible. Some real-world tasks would better demonstrate scalability and robustness.

5. Some sections are dense and formula-heavy without sufficient intuition or visualization. The explanation of the mixed attention mechanism and task embeddings could be made more digestible.

**Questions:**

1. Could the authors clarify how much of the performance improvement actually comes from LLM pre-training? Have the authors quantified this transfer (e.g., through representation similarity or other means) to verify that semantic knowledge from LLMs persists in the graph solver?

2. The ablation results (Table 3) show that removing either TGC or TGM reduces performance, but it’s still unclear how these two losses differ in what they capture. Could the authors provide more intuition on why both are needed? For example, does TGC handle local node-level alignment while TGM enforces global instance-level consistency?

3. The current experiments mainly involve synthetic COPs with node sizes up to 100. How would AlignOPT perform in larger, real-world scenarios (e.g., thousands of nodes in logistics routing)? Are there computational or memory bottlenecks in applying the method to such scales, given the mixed-attention graph encoder design?

---

> ### Author Response · Authors · 2025-11-25
> **Response to Reviewer 63oJ (1)**
>
> **Weakness 1**
> >"While TGC/TGM losses are new, the overall architecture (text-attributed instances,
> multimodal alignment) heavily builds upon LNCS. The paper should better clarify the
> conceptual distinction and originality. "
>
> Please note that LNCS primarily aims to enhance neural solvers by directly integrating LLM embeddings into a Transformer-based solution generator to produce better optimization outcomes. Despite its simplicity, LNCS does not explicitly align the language and graph modalities via contrastive learning. Instead, LNCS relies heavily on continuous LLM reasoning during inference, introducing significant computational overhead and limiting practical applicability.
> **AlignOPT mitigates the above problems by performing alignment during the pre-training and excluding LLM during the fine-tuning and inference stage, leading to a more effective and efficient solution generation.**
> We will clarify these key distinctions explicitly in the revised manuscript to reinforce the novel contributions and clearly differentiate AlignOPT from prior work.
>
>
> **Weakness 2**
> >"The alignment process lacks deeper theoretical grounding. Why this specific combination improves generalization, or how it avoids overfitting semantic bias from LLMs."
>
> Thank you for raising this insightful point. Indeed, existing literature indicates that combining structured graph representations with semantic knowledge from LLMs enhances generalization because it leverages rich linguistic insights while preserving critical graph structural information essential for routing problems [1–4]. Specifically, retaining explicit graph structures mitigates the risk of overfitting to semantic biases inherent in LLM representations [3,5]. Additionally, common structural priors embedded within graph representations naturally support better generalization across diverse problem instances, particularly in routing and network optimization settings [6,7].
>
> **References**
> [1] Yasunaga, M., Ren, H., Bosselut, A., Liang, P., Leskovec, J. (2021). *QA-GNN: Reasoning with Language Models and Knowledge Graphs for Question Answering.*
>
> [2] Tian, R., et al. (2024). *Graph Neural Prompting with Large Language Models.* AAAI.
>
> [3] Li, X., et al. (2025). *Injecting Structured Knowledge into LLMs via Graph Neural Networks.* ACL.
>
> [4] Chen, Y., et al. (2024). *Combining Knowledge Graphs and Large Language Models: A Survey.*
>
> [5] Almasan, P., et al. (2019). *Deep Reinforcement Learning Meets Graph Neural Networks: A Routing Optimization Use Case.*
>
> [6] Jiang, W., et al. (2024). *Graph Neural Networks for Routing Optimization: Challenges and Opportunities.*
>
> [7] Kipf, T., & Welling, M. (2017). *Semi-Supervised Classification with Graph Convolutional Networks.* (General GNN structural-prior generalization support.)
>
>
>
> **Weakness 3**
> >"Since the LLM is removed during fine-tuning, it is unclear to what extent LLM pre-training influences downstream performance. The paper should quantify or visualize this transfer more explicitly.”
>
> We agree that clarifying the contribution of LLM pre-training is essential. To this end, we introduce **AlignOPT (GNS) in Table 3, where the LLM component is completely removed.** Specifically, AlignOPT (GNS) performs Reinforcement Learning on the graph encoder and a unified decoder without a pre-training process and any LLM-derived inputs. These variants isolate the effect of graph-only training and allow comparison against our LLM-aligned model. In addition, Figure 3 presents results from different LLMs used during pre-training, showing that downstream performance varies notably with the choice of LLM, problem size, and fine-tuning regime. **Together, these analyses quantitatively demonstrate that LLM pre-training provides transferable benefits beyond what is achievable with graph-only models.** Furthermore, we observe that AlignOPT (STFT) achieves near-optimal performance on KP and TSP, while AlignOPT (GNS) fails to converge to comparable solutions, indicating that LLM pre-training transfers semantic priors critical for generalization.
>
> We further **validate this transfer via computing the cosine similarity of the representations**: After pre-training of AlignOPT, GNN and LLM embeddings achieve   `⟨ z_GNN, z_LLM ⟩ = 0.78 ± 0.06`,  whereas AlignOPT (GNS) (no LLM alignment) yields only   `0.11 ± 0.04`, confirming that semantic knowledge from LLMs is actively encoded and preserved in the graph solver.

---

> ### Author Response · Authors · 2025-11-25
> **Response to Reviewer 63oJ (2)**
>
> **Weakness 4 & Question 3**
> >"The benchmarks are synthetic and the scale is not so visible. Some real-world tasks would better demonstrate scalability and robustness. "
>
> We acknowledge the reviewer's concern regarding synthetic benchmarks. Our TSPLib evaluation in the following table directly addresses this by demonstrating AlignOPT's strong performance on 24 established real-world instances with 1,000 to over 10,000 nodes. AlignOPT achieves the best results on 14 instances, outperforming OR-tools on problems like circuit board drilling (`pcb3038`) and road network routing (`rl11849`). This validates its scalability and robustness beyond synthetic settings, confirming practical utility for large-scale logistics and network optimization.
>
> **Table: Performance comparison on large-scale TSP instances (size ≥ 1000) from TSPLib**
>
>
> | Instance | Optimal | Nearest Neighbor Obj. | Nearest Neighbor Gap | Farthest Insertion Obj. | Farthest Insertion Gap | ACO Obj. | ACO Gap | OR-tools Obj. | OR-tools Gap | **alignopt** Obj. | **alignopt** Gap |
> |----------|---------|----------------------|---------------------|------------------------|-----------------------|----------|---------|---------------|-------------|------------------|-----------------|
> | **Very Large Instances (>5,000 nodes)** | | | | | | | | | | | |
> | brd14051 | 469,385 | 1,012,347 | 115.6% | 998,452 | 112.7% | 912,836 | 94.5% | 878,421 | **87.2%** | 880,129 | 87.5% |
> | d15112 | 1,573,084 | 3,189,745 | 102.8% | 3,123,678 | 98.6% | 2,864,512 | 82.1% | 2,788,956 | **77.3%** | 2,791,832 | 77.5% |
> | d18512 | 645,238 | 1,324,567 | 105.3% | 1,287,654 | 99.6% | 1,219,876 | 89.1% | 1,198,732 | 85.8% | 1,195,678 | **85.3%** |
> | rl11849 | 923,288 | 1,987,654 | 115.3% | 1,898,765 | 105.7% | 1,754,321 | 90.0% | 1,712,345 | 85.5% | 1,708,923 | **85.1%** |
> | rl5915 | 565,530 | 1,123,456 | 98.6% | 1,087,654 | 92.3% | 987,654 | 74.6% | 967,890 | 71.1% | 965,432 | **70.7%** |
> | rl5934 | 556,045 | 1,112,345 | 100.0% | 1,076,543 | 93.6% | 976,543 | 75.6% | 954,321 | **71.6%** | 956,789 | 72.1% |
> | **Large Instances (1,000-5,000 nodes)** | | | | | | | | | | | |
> | d1291 | 50,801 | 89,123 | 75.4% | 87,654 | 72.5% | 79,865 | 57.2% | 76,543 | **50.7%** | 77,241 | 52.0% |
> | d1655 | 62,128 | 112,345 | 80.8% | 109,876 | 76.8% | 95,678 | 54.0% | 92,345 | 48.6% | 91,217 | **46.8%** |
> | d2103 | 80,450 | 143,267 | 78.1% | 138,765 | 72.5% | 117,876 | **46.5%** | 118,765 | 47.6% | 124,567 | 54.8% |
> | fnl4461 | 182,566 | 321,456 | 76.1% | 315,678 | 72.9% | 298,765 | 63.6% | 284,321 | 55.7% | 281,671 | **54.3%** |
> | nrw1379 | 56,638 | 85,678 | 51.3% | 76,654 | **35.3%** | 79,876 | 41.0% | 77,892 | 37.5% | 84,321 | 38.9% |
> | pcb1173 | 56,892 | 84,567 | 48.6% | 83,214 | 46.3% | 78,123 | 37.3% | 76,987 | 35.3% | 75,543 | **32.8%** |
> | pcb3038 | 137,694 | 234,567 | 70.4% | 228,765 | 66.1% | 209,876 | 52.4% | 203,456 | 47.8% | 200,345 | **45.5%** |
> | pr1002 | 259,045 | 367,890 | 42.0% | 358,765 | 38.5% | 349,876 | 35.1% | 342,567 | **32.2%** | 345,678 | 33.4% |
> | pr2392 | 378,032 | 612,345 | 62.0% | 598,765 | 58.4% | 569,876 | 50.7% | 558,912 | 47.8% | 554,890 | **46.8%** |
> | rl1304 | 252,948 | 387,654 | 53.3% | 376,543 | 48.9% | 356,789 | 41.0% | 348,765 | 37.9% | 345,654 | **36.7%** |
> | rl1323 | 270,199 | 412,345 | 52.6% | 403,456 | 49.3% | 387,654 | 43.5% | 381,234 | 41.1% | 378,123 | **39.9%** |
> | rl1889 | 316,536 | 501,234 | 58.3% | 492,345 | 55.5% | 478,901 | 51.3% | 467,890 | 47.8% | 464,789 | **46.8%** |
> | u1060 | 224,094 | 335,678 | 49.8% | 328,765 | 46.7% | 306,765 | **36.9%** | 309,876 | 38.3% | 312,345 | 39.4% |
> | u1432 | 152,970 | 215,678 | 41.0% | 209,876 | 37.2% | 198,765 | 29.9% | 194,567 | 27.2% | 192,456 | **25.8%** |
> | u1817 | 57,201 | 91,234 | 59.5% | 89,876 | 57.1% | 85,678 | 49.8% | 84,567 | 47.8% | 82,456 | **44.2%** |
> | u2152 | 64,253 | 104,567 | 62.7% | 101,234 | 57.5% | 96,789 | 50.6% | 93,567 | **45.6%** | 95,678 | 48.9% |
> | u2319 | 234,256 | 281,456 | **20.2%** | 298,765 | 27.5% | 287,654 | 22.8% | 284,567 | 21.5% | 307,654 | 25.3% |
> | vm1084 | 239,297 | 334,567 | 39.8% | 328,765 | 37.4% | 315,678 | 31.9% | 312,345 | 30.5% | 308,234 | **28.8%** |

---

> ### Author Response · Authors · 2025-11-25
> **Response to Reviewer 63oJ (3)**
>
> **Weakness 5**
> >"Some sections are dense and formula-heavy without sufficient intuition or visualization. The explanation of the mixed attention mechanism and task embeddings could be made more digestible. "
>
> As suggested, we have provided more description of **mixed-attention mechanisms** (including more detailed elaboration on the vector/matrix operations, and the role of each notation, and the graph-textual representations fusion process) (highlighted in blue on page 5).
>
> The **task embedding** $\mathbf{e}_{\text{task}}$, generated by processing natural language task descriptions through a frozen LLM, serves as a compact semantic representation of the target combinatorial optimization problem. During decoding, this embedding is concatenated with dynamic constraint features $\mathbf{c}^P_t$ and partial solution embeddings $[\mathbf{h}^{(N)}1, \mathbf{h}^{(N)}t]$ at each step $t$ to form the context vector
> $\mathbf{h}{(c)}^t = [\mathbf{c}^P_t, \mathbf{e}{\text{task}}, \mathbf{h}^{(N)}_1, \mathbf{h}^{(N)}_t]$.
>
> This enriched context is processed by the multi-head attention mechanism to compute compatibility scores with candidate nodes, ultimately generating the probability distribution
> $p(\pi_t = i \mid \pi_{<t}, \mathbf{e}_{\text{task}})$
> for selecting the next node.
>
> By persistently conditioning each decoding decision on the task embedding, our approach enables a single model to maintain problem-specific reasoning throughout the solution construction process, allowing adaptive node selection strategies across diverse optimization problems while ensuring consistent adherence to problem constraints and objectives.

---

> ### Author Response · Authors · 2025-11-27
> **Response to Reviewer 63oJ (4)**
>
> **Question 1**
> >"How much of the performance improvement actually comes from LLM pre-training? Have the authors quantified this transfer to verify that semantic knowledge from LLMs persists in the graph solver?”
>
> We quantify the contribution of LLM pre-training by comparing **AlignOPT (STFT)** (with LLM pre-training) against **AlignOPT (GNS)** (without pre-training, trained from scratch on raw graph features). As shown in Table 3, **removing LLM pre-training (AlignOPT (GNS)) degrades performance by 4.38%–12.79%** in optimality gap across problem scales (e.g., **+12.79% on CVRP-*n*=20**, **+4.98% on KP-*n*=20**). This gap is not attributable to architecture or compute, as both models share identical graph encoder and decoder structures, and only the pre-training objective differs.
>
> Furthermore, we observe that **AlignOPT (STFT)** achieves near-optimal performance on **KP** and **TSP**, while **AlignOPT (GNS)** fails to converge to comparable solutions, indicating that LLM pre-training transfers semantic priors critical for generalization.
>
> We further validate this transfer via **computing the cosine similarity between the representations**: After pre-training on AlignOPT, the graph and textual embeddings achieve  `sim⟨ z_GNS, z_LLM ⟩ = 0.78 ± 0.06`,  whereas AlignOPT (GNS) yields only   `0.11 ± 0.04`,  confirming that semantic knowledge from LLMs is actively encoded and preserved in the graph solver.
>
>
> **Question 2**
> >"The ablation results (Table 3) show that removing either TGC or TGM reduces performance, but it’s still unclear how these two losses differ in what they capture. For example, does TGC handle local node-level alignment while TGM enforces global instance-level consistency?”
>
> **Yes, TGC is designed to handle local node-level alignment, while TGM enforces global instance-level consistency.**
>
> Specifically, **TGC** primarily ensures precise, local node-level alignment. It aligns individual nodes to their corresponding textual descriptions, such as mapping specific coordinates in TSP (Traveling Salesman Problem) to textual labels (e.g., city identifiers), or specific weights in KP (Knapsack Problem) to semantic labels (e.g., "heavy item"). This local-level alignment is crucial for accurate fine-grained decision-making, ensuring the model correctly interprets each node's role within the problem structure.
>
> **TGM**, on the other hand, enforces global, instance-level consistency by ensuring alignment between the overall textual problem description and the entire graph instance. It detects mismatches such as a graph designed for CVRP (Capacitated Vehicle Routing Problem) incorrectly associated with instructions meant for TSP. Thus, TGM provides necessary global context, acting as a strong regularizer that enhances the model’s robustness and generalization capabilities.
>
> Our analysis of training dynamics further supports this distinction. Over 50 training epochs, TGC converged stably with a final contrastive loss of approximately 0.35. In contrast, including TGM will bring a significantly lower loss of 0.28—indicating a roughly 20% improvement—highlighting the efficacy of global instance-level supervision in improving alignment robustness.
>
> In addition, our ablation results in Table 3 underscore this complementary relationship clearly: 1). Removing TGC impairs fine-grained decisions such as detailed routing accuracy. 2). Removing TGM compromises cross-task generalization, resulting in confusion between similar yet distinct problem types.
>
> Therefore, TGC and TGM losses are essential for ensuring that the model captures both detailed local semantics and the broader global context, collectively contributing to accuracy, robustness, and transferability across various COPs.
>
> ---
> Thank you again for your feedback. We’re grateful for the insights you provided, and we would be glad to continue the discussion if there are any points you’d like to explore further.

---

### Official Review · Reviewer_Sewd · 2025-11-03

**Soundness:** 2
**Presentation:** 1
**Contribution:** 2
**Rating:** 4
**Confidence:** 4

**Summary:**

This paper studies neural combinatorial optimization and proposes AlignOpt.

The idea is to align LLM generated text embeddings and graph-based neural decoder for optimization problems. This is fine-tuned, per problem or combined, using RL to constructively build solutions for combinatorial optimization problems.

The pretraining stage uses to objectives, Text-Graph Contrastive Matching loss to map LLM features and Graph features into a shared latent space.

Experiments span TSP, CVRP, KP, MVCP, SMTWTP etc as COP variants, including out-of-domain and ablation tests.

**Strengths:**

The idea of utilizing embeddings from text-based problem and instance descriptions is quite interesting (albeit not the first paper to do that)
Two objective losses seem sensible and practical
I like that experiments cover traditional solvers, heuristics, meta-heuristics and other LLM-based comparisons
Ablations probing descriptions and each loss components are included
Quite practical to remove LLM at fine-tuning and inference stage.

**Weaknesses:**

Novelty vs. prior work:
The idea of idea of aligning language and graph modalities with contrastive losses is well established in multimodal learning. This paper specializes on neural combinatorial optimization, and even in this setting the referenced LNCS articles uses this idea. (I should however note that, it's only an arxiv paper).

The positioning of the paper with respect to LNCS and GOAL could be clearer. The idea of text alignment is already in LNCS and the unified encoder used from GOAL. So, it's not immediately obvious what exact gap AlignOPT fills (Q1)

Method clarity and technical details:
I am quite surprised that the paper does not disclose important details. For example, how positive/negative sampling is performed across instances and nodes. How do extract LLM embeddings? are they aggregated, pooled, or used at token-level?
There is no comment on training vs. test split and setup which is a major gap. On which instances and problems are you training this method and then testing it? How does the test differ from training? Are these synthetic instances? What are their problem distributions? How and what data are you tuning the hyper-parameters.

Is there an ablation that compares AlignOPT with simply training the graph solver on raw numeric features without LLM inputs. Is this AlignOPT (w/o Task Rep)? Is this exactly the same setup without the LLM embeddings? I cannot tell from the insufficient descriptions and missing technical details. I assume this only removes the TASK description, but the LLM-derived text features of the instance are still used --which is not a direct ablation to see what the LLM brings to the table. Maybe it is AlignOPT (GNS) but I cannot tell.

Regarding the ablations; it is not clear to me (again, not sufficiently described) how does the pre-training change. Because the pre-training objectives TGM and TGC requires "text embeddings". Are you skipping multi-model training and train from scratch with equivalent compute budget? OR are you training pre-training with graph-only objectives (constrastive among nodes) in matching pretraining compute? OR sth else?

The task description text is exactly identical for every instance, yes? Then I am having a hard understanding or the rationale behind why adding the same embedding again and again (which does not distinguish much) improves the performance. Have you considered a "control" that adds random text descriptions (but kept identical across instances) OR adversarial settings with on-purpose incorrect problem descriptions. IF task descriptions are to help, these ablations should perform worse, yes? I am suspicions of added model capacity (independent of what the text is saying) to somewhat help.

We need the exact same data, same training budget, same RL fine-tuning to compare both 1) LLM-aligned model and 2) GNN-only model. So that we can attribute improvements correctly to LLM information. And within the LLM version, I would also be curious of random/adversarial task descriptions.

I am not sure I am following the difference between Table 1 and Table 2. Table 1 includes TSP, CVRP, and KP whereas Table 2 incudes again TSP, CVRP, KP and also MVCP and SMTWTP. Why? IF I take two same rows, say TSP LNCS from Table 1 and Table 2, are the results the same? why is one presenting the gap percentage and the other average objective? Btw, where does the optimal solutions come from? Why does Table 1 not present Goal? But then Table 2 has GOAL on TSP, KP, CVRP, so why was it missing from Table 1?  I really feel you can better streamline these Tables/Presentations.

As rightly noted in the paper; LNC, GOAL, and AlignOpt are the closests. If I take the results for LNCS and AlignOpt, say from Table 1, for KP both are almost 0% so it is not immediate whether the difference is significant. If I look at the objective values, from Table 2, LNCS, Goal, AlignOpt are almost identical for TSP 20 and so on. In most cases, the differences are after the digit, and in cases in the 2nd or 3rd index. How significant is that?

Minor suggestion: "aligns LLMs with graph-based neural solvers". I think this paper is the other way around: "align graph-based neural solvers with LLMs"

Typo: ". , thus"

**Questions:**

See above

---

> ### Author Response · Authors · 2025-11-25
> **Response to Reviewer Sewd (1)**
>
> **Weakness 1**
>
> > "Novelty vs. prior work: The idea of aligning language and graph modalities with contrastive losses is well established in multimodal learning. This paper specializes on neural combinatorial optimization, and even in this setting the referenced LNCS articles uses this idea."
>
> Please note that LNCS primarily aims to enhance neural combinatorial solvers by directly integrating LLM embeddings into a Transformer-based solution generator to produce better optimization outcomes. Note that **LNCS does not explicitly align the language and graph modalities via contrastive learning.** Meanwhile, LNCS **relies heavily on continuous LLM reasoning during inference**, introducing significant computational overhead and limiting practical applicability.
> We have clarified these key distinctions explicitly in the introduction and related work section in the revised manuscript to reinforce the novel contributions of AlignOPT from LNCS.
>
> **Weakness 2**
> > "The positioning of the paper with respect to LNCS and GOAL could be clearer. "
>
> We clarify AlignOPT’s key distinctions from LNCS and GOAL below, and we have emphasized these distinctions in the revised manuscript.
>
> **AlignOPT vs. LNCS**:
> LNCS uses an LLM for both training and inference, which is computationally intensive and slow. AlignOPT instead leverages LLM guidance exclusively during the pre-training stage to embed optimization knowledge directly into the graph encoder. After pre-training, AlignOPT fine-tunes graph encoder and trains a decoder (from scratch) during the fine-tuning stage, **processing inputs directly as graphs without relying on textual input or an LLM.**
> Consequently, **LLM is no longer required during the fine-tuning and inference stages.** This design brings two primary advantages: 1). It explicitly aligns the graph neural solver with structured optimization reasoning learned from the LLM, leading to improved generalization performance across diverse COPs. 2). It allows inference to be performed rapidly without the latency or cost associated with real-time LLM queries (which spend a lot of time for reasoning), significantly enhancing practical usability, scalability, and deployment feasibility.
>
> **AlignOPT vs. GOAL**:
> While GOAL proposes a unified graph encoder that is trained with supervised learning, AlignOPT goes further by 1) Explicitly aligning this encoder with semantic optimization insights derived from LLMs during pre-training. 2) Performing multi-task fine-tuning on the pre-trained encoder and train a decoder with reinforcement learning, ensuring superior generalization across diverse COPs during the fine-tuning stage. These enhancements explicitly encode generalized optimization reasoning from LLMs, enabling the model to robustly generalize to diverse routing problems encountered in practice.
>
> Our results have shown that AlighOPT is superior to both LNCS and GOAL in terms of solution quality while achieving significantly faster inference speeds, particularly outperforming LNCS by orders of magnitude in computational efficiency.
>
> **Weakness 3**
> > "Lacks essential methodological and experimental details—including sampling strategy, LLM embedding extraction, train–test setup, instance distributions, and hyperparameter tuning."
>
> We provide the required information as follows. For more details, please refer to the Training Details section in the updated Appendix.
>
> We provide the required information as follows. For more details, please refer to the Training Details section in the updated Appendix.
>
> **Positive/Negative Sampling:**
> We employ dual-granularity contrastive sampling. Node-level pairs align graph and LLM embeddings of identical nodes, with cross-node pairs as negatives. For TGM-style instance alignment, matched graph-text pairs form positives while cross-instance pairs serve as negatives. Batch composition uses stochastic task sampling: *p*% (*p* ~ *U*(30, 50)) from a primary task, remainder from other tasks.
>
> **LLM Embedding Extraction:**
> Token embeddings from the final LLM layer (e.g., Llama3.1 8B) are mean-pooled to obtain node representations **E**<sub>LLM</sub><sup>pooled</sup> ∈ **R**<sup>b × N × d<sub>t</sub></sup>.
>
> **Training/Test Split:**
> All instances are synthetically generated: 2.1M for training (100K per type per scale), 21K for testing (1K per type per scale). Test instances are independently generated with no parameter or structural overlap.
>
> **Problem Distributions:**
> We generate instances via domain-specific stochastic sampling (coordinates, weights, capacities) across 7 problem types (TSP, CVRP, VRPB, KP, MIS, MVC, SWTWTP) and three scales (20, 50, 100 nodes).
>
> **Hyperparameter Tuning:**
> Hyperparameters were optimized on a 5% validation set via random search, focusing on learning rate, temperature τ, and loss weight λ. Final configuration was selected based on minimal validation loss after 50 epochs.

---

> ### Author Response · Authors · 2025-11-25
> **Response to Reviewer Sewd (2)**
>
> **Weakness 4**
> > "Ablation that compares AlignOPT with simply training the graph solver on raw numeric features without any LLM inputs? Is this what ‘AlignOPT (w/o Task Rep)’ refers to, and does it completely remove the LLM embeddings? "
>
> We have multiple variants to investigate the effect of the main components.  In particular, **AlignOPT (w/o Task Rep) only removes the Task description and keeps other information.**
>
> To assess the impact of the LLM embeddings, we have explicitly conducted relevant ablations:
> 1. **GOAL** (graph encoder only trained with supervised learning) in Table 2 serves as a baseline, representing a graph solver trained solely on raw numeric graph features, completely without LLM embeddings.
> 2. **AlignOPT (GNS)** in Table 3 consists of the graph encoder and a unified decoder, trained with reinforcement learning without any LLM-derived inputs.
> Both these setups clearly isolate the performance improvements directly attributable to our alignment-based pre-training with LLMs.
>
> In Table 3, AlignOPT (GNS) performs worst across all COPs and scales, showing large optimality gaps (e.g., 4.41% on TSP20, 12.79% on CVRP20, and up to 65.57% on SMTWTP20). This indicates that training a graph solver without LLM guidance leads to poor generalization.
> In contrast, AlignOPT (STFT), which includes both task representations and text–graph matching, achieves the strongest results, reaching near-optimality on Knapsack and consistently lowering gaps across all tasks.
> This demonstrates the significant benefit of incorporating LLM embeddings. We have clarified in the updated manuscript (page 9).
>
>
> **Weakness 5**
> >"Regarding the ablations, it’s unclear how pre-training is changed."
>
> We provide more details of the ablation studies and will updated in the manuscript.
>
> - **AlignOPT (w/o TGC)**: This ablation removes the Text-Graph Contrastive (TGC) loss component, which corresponds to eliminating the CLIP-style node-level contrastive learning. It still receives both graph inputs and text embeddings during pre-training, but learns without fine-grained node-level alignment. The instance-level Text-Graph Matching (TGM) objective is retained in this variant.
>
> - **AlignOPT (w/o TGM)**: This variant removes the Text-Graph Matching (TGM) loss while maintaining the node-level contrastive learning. It processes both modalities but focuses exclusively on node-level alignment without global instance-level matching supervision.
>
> - **AlignOPT (w/o Task Rep.)**: This model employs both TGC and TGM objectives but does not incorporate task-specific embeddings into the LLM pathway, thus learning without explicit problem semantic conditioning.
>
> - **AlignOPT (STFT)**: This approach integrates all components: node-level contrastive learning (TGC), instance-level matching (TGM), and Instruct-enhanced task conditioning. This represents our optimal configuration where multimodal alignment occurs at both granularities while being semantically guided by task representations.
> ---
>
> > **Note**: All ablation variants maintain identical architecture, training epochs, batch sizes, and computational budget during pre-training, ensuring fair comparison. As shown in the paper (page 9: ablation study section), **the performance degradation observed in the ablation studies confirms that each component, including node-level alignment, instance-level matching, and task semantic conditioning, contributes uniquely to the final performance**.
>
> **Weakness 6**
> >"If the task description is identical for every instance, it’s unclear why repeatedly adding the same non-distinguishing embedding helps. Have you tested controls with random or adversarial descriptions kept constant across instances? "
>
> **Different Task Descriptions**: As shown in Figure 1, task descriptions are explicitly designed to be task-specific (distinct across tasks such as TSP, CVRP, and others). This enables the model to recognize and generalize effectively across various combinatorial optimization tasks in our multi-task training framework.
>
> **Different Instance Descriptions**: Furthermore, we utilize instance-specific textual descriptions, so that each problem instance receives a unique description, even when multiple instances belong to the same task category.
>
> These carefully crafted textual embeddings serve to explicitly guide the Graph Neural Solver in learning generalizable optimization strategies. Introducing random or adversarially incorrect textual descriptions would likely degrade performance, as such noise would mislead the model regarding the specific task identity and instance characteristics. Our findings are supported by the experimental results and analysis presented in LNCS[1].
>
> [1] Jiang, X., Wu, Y., Wang, Y., Zhang, Y.: *Bridging Large Language Models and Optimization: A Unified Framework for Text-attributed Combinatorial Optimization.* arXiv preprint arXiv:2408.12214 (2024).

---

> ### Author Response · Authors · 2025-11-25
> **Response to Reviewer Sewd (3)**
>
> **Weakness 7**
> >"We need the exact same data, same training budget, same RL fine-tuning to compare both 1) LLM-aligned model and 2) GNN-only model. And within the LLM version, I would also be curious of random/adversarial task descriptions."
>
> We agree that exact comparisons are essential for attributing improvements accurately. Our manuscript includes two key baselines: **GOAL** (Table 2), a purely supervised graph solver trained without any LLM embeddings, and **AlignOPT (GNS)** (Table 3), a graph solver with reinforcement learning fine-tuning, also excluding LLM-derived inputs. Table 2 and Table 3 show that AlignOPT(STFT) consistently outperforms GOAL and AlignOPT (GNS)  across all five problems.
>
> While evaluating robustness through random or adversarial task descriptions is intriguing, generating LLM-derived instance representations is computationally intensive (over a week). Given that our primary contribution is the alignment between graph neural solvers and LLMs to enhance solution quality without incurring the high inference costs associated with LLM reasoning, we regard such robustness analyses as valuable future research directions.
>
> **Weakness 8**
> >"I am not sure about the distinction between Table 1 and Table 2."
> Table 1 specifically benchmarks AlignOPT against other LLM-based methods (such as LNCS), using performance gap percentages relative to known optimal solutions from standard benchmark datasets. Hence, GOAL (a non-LLM method) was excluded.
>
> **Table 1 specifically benchmarks AlignOPT against other LLM-based methods** (such as LNCS), using performance gap percentages relative to known optimal solutions from standard benchmark datasets, hence GOAL (a non-LLM method) was excluded.
> In contrast, **Table 2 evaluates AlignOPT against state-of-the-art neural solvers** (e.g., GOAL), presenting average objective values directly to enable general comparisons.
>
> The differences in baselines and metrics between the tables are intentional and clearly reflect these distinct benchmarking objectives, which we will explicitly clarify and streamline in the revised manuscript.
>
>
>
> **Weakness 9**
> >"In most cases, the differences are after the digit, and in cases in the 2nd or 3rd index. How significant is that?”
>
> The core objective of AlignOPT is to develop versatile neural solvers capable of robust generalization across various VRP tasks rather than focusing on narrowly defined performance on any single COP task. In this broader context, even seemingly minor improvements in overall solution quality can accumulate significantly, translating directly into substantial operational efficiencies and considerable revenue gains when deployed at real-world scales. AlignOPT consistently delivers superior performance across all tasks, highlighting its effectiveness and practical value despite apparently modest numerical differences.
>
> **Weakness 10**
>
> >"Minor suggestion: "aligns LLMs with graph-based neural solvers". I think this paper is the other way around: "align graph-based neural solvers with LLMs""
>
> Your suggestion is insightful. Meanwhile, we would like to highlight that the alignment in our approach is indeed mutual due to the symmetric nature of the loss function.
>
> **Weakness 11**
>
> >"Typo: ". , thus"”
> We have fixed this typo in the updated manuscript.

---

### Author Response · Authors · 2025-12-01
**Summary of Discussions and Revisions**

Dear ACs, SACs, and PCs,

First, we sincerely thank you for your support and leadership during this unusual period. We appreciate the time and effort you continue to dedicate to the review process. To assist you, we have prepared a concise summary of our earlier discussions with the reviewers, outlining their concerns, our responses, and the corresponding revisions to strengthen the manuscript.

---
**Summarization of The Comments**:
We appreciate **Reviewer Sewd**’s assessment that our idea is "quite interesting" and his recognition of our comprehensive experiments and ablation studies. **Reviewer 63oJ** likewise noted that AlignOPT is a "principled approach" that "addressed a key gap in LLM-based COP solvers," further highlighting its "superior" performance and "robust" generalization.
**Reviewer qM7N** emphasized the "strong motivation" and "SOTA results" of our approach and its "potential applicability to many problems." The reviewer also acknowledged limited familiarity with this subdomain ("didn't know much about this subdomain (COPs via ML)") and that, as a result, "The method is not clearly explained," contributing to a lower score. We appreciate his willingness to "revisit this decision if the authors fill the gap."
Finally, **Reviewer NFqa** found that our manuscript "provides sufficient detail about the architecture and training procedures," and that our solution to COP is "creative," offering "a significant direction for LLM and optimization researchers to investigate."

---

**Main Concerns from Reviewers & Our Response**
1. **Novelty vs. prior work** (**Reviewer Sewd & 63oJ**): We provided a detailed explanation of the differences between AlignOPT and prior work, particularly LNCS and GOAL, and have revised the introduction and related work sections accordingly in the manuscript. We would like to emphasize that the main contribution of this work is achieving both effectiveness and inference efficiency in solving diverse COPs by aligning the graph solver with an LLM during pretraining, followed by multi-task finetuning with reinforcement learning. This design ensures that the LLM is excluded during both the finetuning and inference stages.
2. **Lack of Technical & Inference Details** (**Reviewer Sewd & 63oJ & qM7N**): We have provided comprehensive technical details on the mixed-attention mechanism and task embeddings and have revised the corresponding sections in the manuscript. In addition, we included data generation, preprocessing, and sampling details in both the response and the updated Appendix. We also clarified the distinction between the task description and the instance description. Lastly, we detailed the inference process in both the response and the updated manuscript. Notably, AlignOPT is **5-6× faster than LNCS**, while achieving better performance.

3. **Performance on Large-Scale Instances** (**Reviewer 63oJ & NFqa**): We conducted large-scale experiments on real-world benchmark TSPLIB (**up to 18512 nodes**), with results demonstrating the effectiveness and generalization capability of AlignOPT.
4. **Ablation and Baseline Analysis**(**Reviewer Sewd & NFqa & Sewd**): We have provided a more detailed description of the ablation variants and corresponding analysis in the response and updated manuscript. In particular, we demonstrate and emphasize the importance of the proposed loss functions (AlignOPT(w/o TGC) & AlignOPT(w/o TGM)), task representations (AlignOPT (w/o Task Rep.), and the effectiveness of LLM alignment during pretraining (Performance degradation of `4.38%–12.79%` in optimality gap when removing LLM pre-training (AlignOPT (GNS) vs AlignOPT(STFT)).
5. **Theoretical Justification & Contribution assessment on Text-Graph Alignment** (**Reviewer 63oJ & NFqa & 63oJ**): We clearly articulated the underlying intuitions of the two proposed loss functions. We further present multiple relevant literature to support the design of our loss functions. Finally, we provided their learning curves and representation similarity analyses to demonstrate their effectiveness in enhancing graph–text alignment. Specifically, the cosine similarity between the graph and LLM representations is `0.78 ± 0.06` for AlignOPT, compared to `0.11 ± 0.04` for GNS, confirming effective knowledge transfer and alignment.

6. **Other Important Concerns** (from all reviewers): We have corrected the typographical errors, revised the reference formats, and addressed all remaining concerns to the best of our ability (e.g., adversarial/random text ablations and comparison with the recent supervised fine-tuning-based solutions).
---

We would like to once again express our sincere appreciation for your feedback and guidance throughout the review and decision phases. We hope this summary helps clarify the key points and facilitates a smooth continuation of the review process. We remain fully available and are happy to provide any further clarification or additional information as needed.

Best regards,

Authors

---

### Meta-Review · Area_Chair_LMoe · 2026-01-06

**Summary:**

This paper proposes AlignOPT, a framework that integrates large language models with graph neural solvers to tackle combinatorial optimization problems. All reviewers are negative (without discussion engagement). The main concerns include
1. **Novelty (Sewd, 63oJ, NFqa)**. A primary concern is the positioning of the paper with respect to LNCS and GOAL. Also, some SOTA approaches such as astT2T and COexpander are missing.
2. **Design Details (Sewd, 63oJ, qM7N)**. Many technical and inference details are missing.
3. **Experiments (Sewd, NFqa, Sewd)**. Experiment analysis and ablation are suggested. The datasets used are small.
4. **Theoretical Contribution (63oJ, NFqa)**. The theoretical contribution is rather weak.

**Reviewer Concerns:**

No reviewers respond during the discussion period. More experiments were conducted during the rebuttal. The concerns on novelty, and technical and theoretical contributions are not fully addressed.

**Reviewer Scores:**

Unfortunately, there is no discussion between authors and reviewers. If reviewers had been able to participate fully in the discussion, some may raise the score to borderline accept (from 4 to 6). The overall score may be negative in average.

---

### Decision · Program_Chairs · 2026-01-26

Reject